# Physics-informed Reduced Order Modeling of Time-dependent PDEs via Differentiable Solvers

**Nima Hosseini Dashtbayaz**[*]
Department of Computer Science
University of Western Ontario
London, Ontario, Canada
`nhosse5@uwo.ca`

**Hesam Salehipour**[†]
Autodesk Research
Toronto, Ontario, Canada
`hesam.salehipour@autodesk.com`

**Adrian Butscher**
Autodesk Research
Toronto, Ontario, Canada
`adrian.butscher@autodesk.com`

**Nigel Morris**
Autodesk Research
Toronto, Ontario, Canada
`nigel.morris@autodesk.com`

## Abstract

Reduced-order modeling (ROM) of time-dependent and parameterized differential equations aims to accelerate the simulation of complex high-dimensional systems by learning a compact latent manifold representation that captures the characteristics of the solution fields and their time-dependent dynamics. Although high-fidelity numerical solvers generate the training datasets, they have thus far been excluded from the training process, causing the learned latent dynamics to drift away from the discretized governing physics. This mismatch often limits generalization and forecasting capabilities. In this work, we propose **Ph**ysics-**i**nformed **ROM** ($\Phi$-ROM) by incorporating differentiable PDE solvers into the training procedure. Specifically, the latent space dynamics and its dependence on PDE parameters are shaped directly by the governing physics encoded in the solver, ensuring a strong correspondence between the full and reduced systems. Our model outperforms state-of-the-art data-driven ROMs and other physics-informed strategies by accurately generalizing to new dynamics arising from unseen parameters, enabling long-term forecasting beyond the training horizon, maintaining continuity in both time and space, and reducing the data cost. Furthermore, $\Phi$-ROM learns to recover and forecast the solution fields even when trained or evaluated with sparse and irregular observations of the fields, providing a flexible framework for field reconstruction and data assimilation. We demonstrate the framework's robustness across various PDE solvers and highlight its broad applicability by providing an open-source JAX implementation that is readily extensible to other PDE systems and differentiable solvers, available at `https://phi-rom.github.io`.

## 1 Introduction

Many-query problems in engineering, such as design optimization, optimal control, and inverse problems, rely on exploring the solution manifold of a governing PDE in a large parameter space. Reduced order modeling (ROM) provides an appealing alternative to computationally expensive high-fidelity numerical simulations [1, 2]. Consider an autonomous time-dependent PDE of the form

$$\dot{u} = \mathcal{N}(u; \beta), \quad u(t,x): \mathcal{T} \times \Omega \to \mathbb{R}^m, \tag{1}$$

---

[*]This study was conducted during the author's internship at Autodesk.

[†]Corresponding author

39th Conference on Neural Information Processing Systems (NeurIPS 2025).

where $u$ is an $m$-dimensional vector field of interest, $\dot{u}$ is its time derivative, $\mathcal{N}$ is a nonlinear differential operator in space (possibly parameterized by some parameters $\beta$), and $\mathcal{T}$ and $\Omega \in \mathbb{R}^d$ are the temporal and spatial domains, respectively. When accompanied by appropriate boundary conditions, the goal is to evolve an initial condition $u(0, x)$ to any time $t > 0$. While full-order models discretize $\Omega$ to a grid of $N$ coordinates and solve the resulting high-dimensional system of ODEs, ROMs aim to find a manifold in a low-dimensional space $\mathbb{R}^k$ with $k \ll N$ that captures the characteristics of the solution fields and their temporal dynamics. While the computational cost of full-order models grows rapidly with the spatial dimension $d$ and the complexity of the PDE, ROMs provide a computationally efficient alternative by working in a space with a prescribed low-dimensional structure, allowing for real-time simulations.

Traditionally, projection-based ROMs find a *linear* subspace via linear dimensionality reduction techniques (e.g. Principal Component Analysis or PCA) [3], while more recently, *nonlinear* manifold ROMs use neural networks (e.g. auto-encoders) to learn a compact latent representation $\alpha \in \mathbb{R}^k$ to better capture the nonlinear dynamics of the system [4]. Among the latter, most works utilize the temporal structure of the latent space for temporal evolution of the solution fields by either finding the governing system(s) of ODEs in the latent space through sparse symbolic regression [5–7], or learning a separate neural network to realize the latent dynamics through a recurrent neural network [8–10], a transformer network [11] or a Neural ODE [12–15].

In all the aforementioned works, the latent space is constructed in a data-driven manner, i.e. a numerical solver first generates a large dataset of solution trajectories $u(t, x)$ with different initial conditions and/or parameters for some time window up to time $T$, and then, models are trained on the generated data to learn the latent space and its evolution up to $T$. Nevertheless, the nonlinear manifold obtained by the data-driven training is not guaranteed to be consistent with the true physical dynamics of the system and often fails to generalize to new parameters or unseen dynamics. In particular, it is common to observe growing inaccuracies over time due to error accumulation [10, 16] where the model fails to forecast beyond the time horizon of the training dataset.

In order to enhance data-driven ROMs to be more faithful to the governing conservation laws, (i.e. to render them "physics-informed"), two categorically different methods have been proposed in the literature thus far; (i) augmenting the "PINN" loss [17] to the loss function during the offline training needed for learning the latent space [18–20] and (ii) evaluating the solutions directly in the full physical space during inference based on the exact PDEs as evaluated on a *sub-sampled* spatial grid. This second approach was recently proposed by Chen et al. [21] in which reduction is achieved only during inference as a result of spatial sub-sampling and not directly due to the reduced space of the learned manifold. Both methods leverage Implicit Neural Representations (INRs) and automatic differentiation (AD) to formulate the PDEs used in either offline training or online inference. In all these previously published works, however, the numerical solver, which contains the true governing physics of the system, as discretized, is discarded after the data generation and is left out of the training process. See Appendix A for an extended discussion of the related works.

In this paper, we propose a novel category of physics-informed ROMs, namely one in which the conservation laws are embedded within the training procedure through a differentiable solver. We propose a **Ph**ysics-**i**nformed **R**educed-**O**rder **M**odel ($\Phi$-ROM) that imposes the true physical dynamics of the system on the latent space by learning the latent dynamics directly from the numerical solver. We build on the recent DINo framework introduced by Yin et al. [13] that consists of a conditional INR decoder $D$, which maps latent coordinates $\alpha$ to reconstructed fields, and a latent dynamics network $\Psi(\alpha) = \dot{\alpha}$, which models the temporal dynamics of the latent space, making the framework time and space continuous (i.e. mesh-free). DINo was shown to be effective compared to other neural PDE surrogates. Taking inspiration from hyper-reduction techniques in computational physics [22], we replace the numerical integration of $\Psi$ during training in DINo with direct training of $\dot{\alpha}$ with $\dot{u}$ given by the PDE solver. The feedback from the differentiable solver results in a well-structured latent space and a regularized decoder that is consistent with the true physical dynamics of the system and generalizes better to unseen dynamics induced by new parameters and initial conditions. Taking advantage of the mesh-free decoder, $\Phi$-ROM also learns from irregular and sparse observations of the solution fields, and recovers the full dynamics in the sparse training setting.

In summary, our contributions are as follows. **(1)** We introduce the novel $\Phi$-ROM framework that efficiently incorporates differentiable numerical solvers into the training process of nonlinear ROMs. **(2)** We show that $\Phi$-ROM generalizes better to new parameters and initial conditions, and extrapolates

beyond the training temporal horizon compared to purely data-driven training. **(3)** We demonstrate the effectiveness of $\Phi$-ROM compared to other physics-informed training methods for nonlinear ROMs. **(4)** We further show how $\Phi$-ROM can be trained with sparse observations of the solution fields while maintaining its accuracy, providing a robust framework for data assimilation. **(5)** Finally, we show that $\Phi$-ROM is robust to the underlying numerical method of the solver and its discretization scheme, and can be easily extended to other PDEs and differentiable solvers using our open-source codebase in JAX.

## 2 Background

### 2.1 Problem setting

Consider the PDEs described in Eq. (1). Given an initial condition $u(0, x)$ and parameters $\beta$, we are interested in evolving the PDE in time to find the state $u(t, x)$ for any time $t \in \mathcal{T}$. To this end, $u$ is discretized on a grid $\mathcal{X}$ of $N$ spatial locations in $\Omega$ at some time steps $t \in [0, T]$, creating snapshots $\mathbf{u}^t$ that form a trajectory $\mathbf{u}^{0:T} = \{\mathbf{u}^0, \mathbf{u}^1, \dots, \mathbf{u}^T\}$. A dataset $\mathbf{U}$ then consists of $M$ trajectories $\mathbf{u}_i^{0:T}$, where each trajectory is associated with a different parameter $\beta$ and/or initial condition. In our experiments, we train the models on $[0, T_{tr}]$ sub-interval of each trajectory in the training dataset $\mathbf{U}_{tr}$, and evaluate the models on $[0, T_{tr}]$ (interpolation) and $[T_{tr}, T_{te}]$ (extrapolation) intervals of the trajectories in a test dataset $\mathbf{U}_{te}$ consisting of unseen initial conditions or parameters. Note that we do not make any assumptions regarding the spatiotemporal discretization method or its consistency across the samples during training and inference.

### 2.2 Auto-decoder for continuous spatial reduction

To have a mesh-free spatial reduction pipeline, we adopt the auto-decoding scheme used in [13], where the encoder network is replaced with an inversion step [23], and the decoder is a conditional INR [24]. For a decoder network $D_\theta$ parameterized by $\theta$ and any snapshot $\mathbf{u}$ on a grid $\mathcal{X}$, the goal is to learn a compact latent manifold with coordinates $\alpha \in \mathbb{R}^k$ such that

$$\hat{\mathbf{u}} = D_\theta(\alpha, \mathcal{X}), \tag{2}$$

where $\hat{\mathbf{u}}$ is an approximate reconstruction of $\mathbf{u}$. We drop $\mathcal{X}$ from $D_\theta$ for brevity if there is no ambiguity. Note that $D_\theta$ reconstructs a field at each coordinate $x \in \mathcal{X}$ separately and stacks them to form the full reconstructed field $\hat{\mathbf{u}}$ (see Fig. 5 in Appendix). During training, for each training snapshot $\mathbf{u}_i^t \in \mathbf{U}_{tr}$, a latent vector $\alpha_i^t$ is initialized with zeros and is optimized along with the decoder parameters $\theta$ by minimizing the reconstruction loss $L_{rec}$:

$$\theta, \Gamma = \underset{\theta, \Gamma}{\operatorname{argmin}} \sum_{\substack{(\mathbf{u}_i^t, \alpha_i^t) \\ \in (\mathbf{U}_{tr}, \Gamma)}} L_{rec}(\mathbf{u}_i^t; \theta, \alpha_i^t),$$

$$L_{rec}(\mathbf{u}_i^t; \theta, \alpha_i^t) = \ell(D_\theta(\alpha_i^t), \mathbf{u}_i^t), \tag{3}$$

where $\Gamma$ is the set of all training latent vectors (i.e. $\Gamma = \{\alpha_i^{0:T_{tr}}\}_{i \in [1, M_{tr}]}$), and $\ell$ is a measure of error. We use a normalized error for $\ell$ as defined in Appendix B.3. When trained, the latent coordinates corresponding to a new state are computed by inversion:

$$\hat{\alpha} = D_\theta^\dagger(\mathbf{u}, \mathcal{X}) = \underset{\alpha}{\operatorname{argmin}} L_{rec}(\mathbf{u}; \theta, \alpha), \tag{4}$$

where $D_\theta^\dagger$ is the pseudo-inverse of $D_\theta$ (replacing the traditional encoders in auto-encoders). The minimization problem in Eq. (4) is solved iteratively using a gradient-based optimizer. Note that inversion is only performed once during inference when forecasting an initial condition. See Appendix C for a detailed description of the inversion step during inference.

### 2.3 Reduced-ordered temporal evolution

The spatial reduction described in Section 2.2 forms a low-dimensional latent manifold that captures the spatial information of the high-dimensional physical space. Similar to previous works, we realize the temporal structure of the latent manifold by training another neural network. This *dynamics*

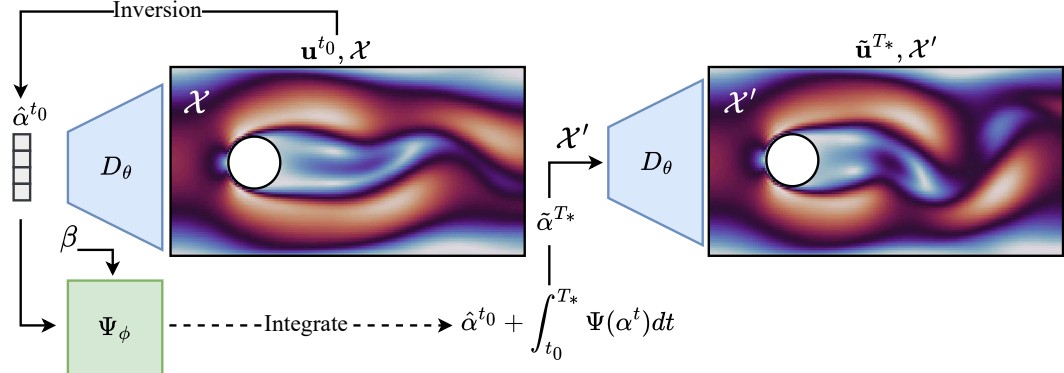

Figure 1: Inference with Φ-ROM. The initial condition and the forecast solution at a target time $T_*$ can be observed and reconstructed on arbitrary (and possibly different) grids $\mathcal{X}$ and $\mathcal{X}'$.

*network* learns the temporal evolution in the latent space, which in turn can be used for time-stepping in $\mathbb{R}^k$ [12, 13]. The dynamics network, $\Psi_\phi$, is therefore defined such that for any latent coordinate $\alpha$,

$$\Psi_\phi(\alpha) = \dot{\alpha} = \frac{d\alpha}{dt}. \tag{5}$$

When trained, the inference involves evolving an initial state $\mathbf{u}^{t_0}$ (discretized on some grid $\mathcal{X}$) up to a desired time $T_*$. For that purpose, first the inversion in Eq. (4) is solved to find the corresponding $\hat{\alpha}^{t_0}$, and then the dynamics network is integrated in time by a numerical ODE solver to find $\tilde{\alpha}^{T_*}$:

$$\tilde{\alpha}^{t_0} = \hat{\alpha}^{t_0} = D_\theta^\dagger(\mathbf{u}^{t_0}, \mathcal{X}) \qquad \tilde{\alpha}^{T_*} = \tilde{\alpha}^{t_0} + \int_{t_0}^{T_*} \Psi(\tilde{\alpha}^t)dt \tag{6}$$

With $\tilde{\alpha}^{T_*}$, the predicted state of the PDE may be reconstructed as $\tilde{\mathbf{u}}^{T_*} = D_\theta(\tilde{\alpha}^{T_*}, \mathcal{X}')$ on a (possibly different) grid $\mathcal{X}'$. This formulation of ROM has been used before in [12, 13, 19], and as will be discussed in Section 3, we build on the same framework in this paper.

*Remark* 2.1. **Parameterized dynamics network** Lee and Parish [25] introduced a parameterized variant of $\Psi$, where the parameter $\beta$ of the PDE (e.g. Reynolds number, diffusivity, etc.) is concatenated with $\alpha$ before being fed into the dynamics network, allowing the dynamics network to learn the temporal evolution of the latent space conditioned on the PDE parameter, i.e. $\dot{\alpha} = \Psi(\alpha, \beta)$. We found that applying a trainable linear transformation to $\beta$ before concatenation with $\alpha$ significantly improves its performance and generalization across the parameters. As such, we adopt this approach in this work whenever the PDE is parameterized (see Appendix E for the detailed architecture). Henceforth, we drop $\beta$ from the notation for brevity.

## 2.4 Differentiable solvers

We define a differentiable solver as a discrete and differentiable nonlinear operator $\mathcal{S}$. For time-dependent PDEs with parameters $\beta$, $\mathcal{S}$ evolves the state $\mathbf{u}$ on grid $\mathcal{X}_\mathcal{S}$ such that $\mathbf{u}^{t+1} = \mathcal{S}[\mathbf{u}^t, \beta, \mathcal{X}_\mathcal{S}]$. We may use $\mathcal{S}$ to obtain $d\mathbf{u}/dt$ which represents the temporal behaviour of the discretized PDE, analogous to its continuous counterpart $\mathcal{N}$ introduced in Eq. (1). See Appendix B.4 for further details on deriving $d\mathbf{u}/dt$ from $\mathcal{S}$.

When $\mathcal{S}$ is implemented within a differentiable framework such as JAX or PyTorch, it can be seamlessly integrated with other neural network components as part of a unified architecture. This integration allows gradients computed during backpropagation to flow directly through the solver. Specifically, during training, we require the derivative of $\mathcal{S}$ with respect to the network parameters $\theta$, denoted $\partial\mathcal{S}/\partial\theta$. This is computed using AD and the chain rule as $\partial\mathcal{S}/\partial\theta = (\partial\mathcal{S}/\partial\hat{\mathbf{u}})(\partial\hat{\mathbf{u}}/\partial\theta)$ where $\hat{\mathbf{u}}$ is the reconstructed field as defined in Eq. (2). Importantly, the term $\partial\mathcal{S}/\partial\hat{\mathbf{u}}$ captures key sensitivity information of the governing PDEs with respect to the state variables–information that is often used in constructing *adjoint* equations for PDE-constrained optimization.

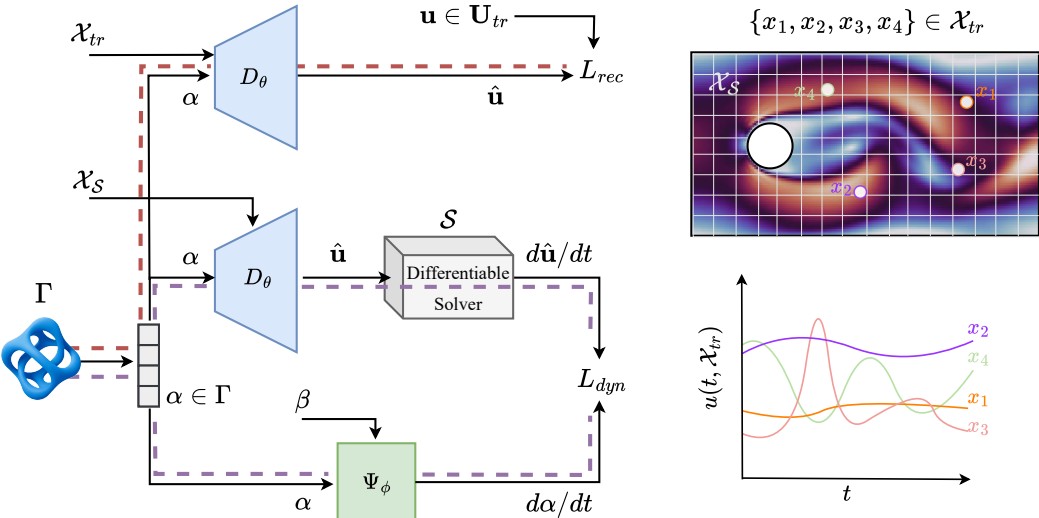

Figure 2: (Left) Training $\Phi$-ROM with the reconstruction loss and dynamics loss. Red and purple dashed lines show the gradient path taken by $L_{rec}$ and $L_{dyn}$. The reconstruction loss trains $D_\theta$ and $\alpha$ with spatial information, while the dynamics loss trains the $\Psi_\phi$ and regularizes the decoder and solution manifolds $\Gamma$ with physics information from $\mathcal{S}$. (Right) $\Phi$-ROM can be trained with partial and irregular observations of the solution fields and recover the full fields and dynamics.

## 3 Physics-informed training with a differentiable solver

### 3.1 Training objective

$\Phi$-ROM is trained to preserve the spatial information of the physical space by minimizing the reconstruction loss $L_{rec}$ in Eq. (3), and to learn the temporal dynamics within the latent space by minimizing a physics-informed dynamics loss $L_{dyn}$.

**Dynamics loss.** By taking the time derivative of the decoder in Eq. (2), we get

$$J_D(\alpha)\dot{\alpha} = \frac{d\hat{\mathbf{u}}}{dt}, \tag{7}$$

where $J_D(\alpha)$ is the Jacobian of the decoder w.r.t. $\alpha$. The true time derivative of the reconstructed state $\hat{\mathbf{u}}$ is given by the numerical solver $\mathcal{S}$ as explained in Appendix B.4. Ideally, the reconstructed dynamics represented by $J_D(\alpha)\dot{\alpha}$ should follow the dynamics given by the solver. However, rather than forming the loss function as $\ell(J_D(\alpha)\dot{\alpha}, d\hat{\mathbf{u}}/dt)$ in the full physical space $\mathbb{R}^{N \times m}$, we project Eq. (7) into the latent space $\mathbb{R}^k$ with $k \ll N$ by taking the pseudo-inverse to get

$$\dot{\alpha}^* = J_D^\dagger(\alpha)\frac{d\hat{\mathbf{u}}}{dt}, \tag{8}$$

where $J_D^\dagger(\alpha)$ is the pseudo-inverse of the Jacobian matrix. In other words, we essentially solve the resulting least-squares problem for $\dot{\alpha}$ in Eq. (7) to find $\dot{\alpha}^*$ as defined in Eq. (8). We will elaborate on the computation of $\dot{\alpha}^*$ via hyper-reduction in Section 3.3. Finally, since the latent dynamics $\dot{\alpha}$ is realized by the dynamics network $\Psi_\phi$, we define the dynamics loss as:

$$L_{dyn}(\alpha, \theta, \phi) = \ell(\Psi_\phi(\alpha), \dot{\alpha}^*). \tag{9}$$

**Training $\Phi$-ROM.** With the reconstruction loss from Eq. (3) and the new dynamics loss in Eq. (9), we define the training objective for $\Phi$-ROM as

$$L_{\Phi\text{-ROM}}(\mathbf{u}; \theta, \alpha, \phi) = \lambda L_{rec}(\mathbf{u}; \theta, \alpha) + (1 - \lambda)L_{dyn}(\theta, \alpha, \phi) \tag{10}$$

minimized over all snapshots $\mathbf{u}_i^t \in \mathbf{U}_{tr}$:

$$\theta, \Gamma, \phi = \underset{\theta, \Gamma, \phi}{\operatorname{argmin}} \frac{1}{|\mathbf{U}_{tr}|} \sum_{\substack{(\mathbf{u}_i^t, \alpha_i^t) \\ \in (\mathbf{U}_{tr}, \Gamma)}} L_{\Phi\text{-ROM}}(\mathbf{u}_i^t; \theta, \alpha_i^t, \phi). \tag{11}$$

In Eq. (10), $\lambda$ controls the regularization effect of the dynamics loss. Choice of $\lambda$ depends on the sensitivity of $\mathcal{S}$ to the noise and errors in the reconstructed fields. We chose $\lambda = 0.5 - 0.8$ depending on the solvers (see Appendix F.3 and G for the choice of $\lambda$). See Appendix B for the detailed training procedure.

*Remark* 3.1. Note that $\dot{\alpha}^*$ and $\hat{\mathbf{u}}$ in Eq. (8) are both functions of $\theta$ and $\alpha$. Since $\mathcal{S}$ is differentiable, gradients of the dynamics loss w.r.t. $\alpha$ and $\theta$ are backpropagated through the solver as discussed earlier in Section 2.4 (see also Fig. 2). As a result, $L_{dyn}$ has a regularizing effect on the decoder and the latent manifold, aligning them with the true dynamics of the solver. This is in contrast with [13], where the decoder and its resulting manifold are trained solely by $L_{rec}$ and are not influenced by the dynamics network.

*Remark* 3.2. One can alternatively obtain the derivatives $d\hat{\mathbf{u}}/dt$ by finding the right-hand side of the exact PDEs using AD similar to PINNs (PINN-ROM) (see [19] for a closely related technique).

## 3.2 Training with data on irregular grids

As pointed out in Section 2.1 and shown schematically in Fig. 2, we do not make any assumptions about the spatial sampling grid of the data and its consistency across snapshots. However, the solver $\mathcal{S}$ may require its input fields to be on a specific grid, $\mathcal{X}_{\mathcal{S}}$, while the training data may be on a different (possibly irregular) grid, $\mathcal{X}_{tr}$. Since $\Phi$-ROM uses an INR decoder, we can still reconstruct the fields on $\mathcal{X}_{\mathcal{S}}$ to compute the dynamics loss, while the reconstruction loss is computed on $\mathcal{X}_{tr}$. In Section 4.4, we will demonstrate that $\Phi$-ROM can be trained reliably with such irregular data sampled at a limited number of sparse locations and recovers the full fields successfully. This ability is particularly helpful for field reconstruction and data assimilation tasks.

## 3.3 Hyper-reduction for computing the dynamics loss

In order to solve for $\dot{\alpha}^*$ defined in Eq. (8), we use QR decomposition to solve the overdetermined system of equations given in Eq. (7) in the least-squares sense. Note that for an $m$-dimensional state reconstruction $\hat{\mathbf{u}} \in \mathbb{R}^{N \times m}$ on a grid $\mathcal{X}_{\mathcal{S}}$ with $N$ coordinates, the Jacobian $J_D(\alpha)$ is of size $N \times m \times k$, whereas $\dot{\alpha} \in \mathbb{R}^k$, where again $k \ll N$ is the reduced latent dimension. While QR decomposition enables solving such an overdetermined system of equations, the computational cost of forming the Jacobian and solving the system grows with $N$ (and the spatial dimension $d$). To mitigate the computational cost, we adopt the *hyper-reduction* technique (see e.g. [22]) in which a subset $\mathcal{X}_{\mathcal{S}}^{\downarrow}$ of $\mathcal{X}_{\mathcal{S}}$ with size $\gamma N$ is sub-sampled for each training snapshot in order to solve the least-squares problem only on the reduced $\mathcal{X}_{\mathcal{S}}^{\downarrow}$. In this work, stochastic hyper-reduction with $\gamma = 0.1$ (as opposed to other techniques such as greedy algorithms) proved to be sufficient for this purpose in all our experiments. Furthermore, we use forward-mode AD to compute the Jacobian matrix. See Appendix B.2 for a detailed description of the dynamics loss computation.

# 4 Results

## 4.1 Experimental settings

We evaluate $\Phi$-ROM on five problems: 1D viscous Burgers' (**Burgers'**) and 2D diffusion (**Diffusion**) equations, both solved by finite-difference solvers, 2D Korteweg–De Vries (**KdV**) equation from the spectral solver Exponax [26], 2D Navier-Stokes decaying turbulence problem (**N-S**) with the finite volume-based solver JAX-CFD [27], and the 2D flow over a cylinder (**LBM**) using the Lattice Boltzmann solver XLB [28]. This diverse set of PDEs and solvers, spanning fundamentally different numerical methods, highlights the robustness and versatility of $\Phi$-ROM. See Appendix D for detailed descriptions of each problem.

For each problem, models are trained on $M_{tr}$ trajectories of $T_{tr}$ time steps and evaluated on $M_{te}$ trajectories to assess forecasting and generalization performance. Evaluation is performed by forecasting solutions for unseen initial conditions or parameters in $\mathbf{U}_{te}$ for $T_{tr}$ (interpolation) and $T_{te}$ (extrapolation) time steps. Among the problems, Burgers' and LBM are parameterized by their source term and Reynolds number, respectively, while the others vary in their initial conditions. Please refer to Appendix F for a detailed description of our experimental settings and Appendix H for extended results.

Table 1: Diffusion and Burgers' forecasting errors (in RNMSEs) of Φ-ROM, DINo, and other physics-informed ROMs.

| | Diffusion | | Burgers' | |
| --- | --- | --- | --- | --- |
| Time | $[0, T_{tr}]$ | $[T_{tr}, T_{te}]$ | $[0, T_{tr}]$ | $[T_{tr}, T_{te}]$ |
| Φ-ROM | **0.080** | **0.034** | 0.021 | **0.028** |
| DINo | 0.089 | 0.051 | 0.021 | 0.060 |
| PINN-ROM | 0.081 | 0.042 | 0.088 | 0.348 |
| FD-CROM | 0.131 | 0.351 | **0.001** | 0.044 |
| AD-CROM | 0.093 | 0.106 | 0.121 | 0.196 |
| ↓AD-CROM | 0.456 | 0.856 | 0.090 | 0.212 |

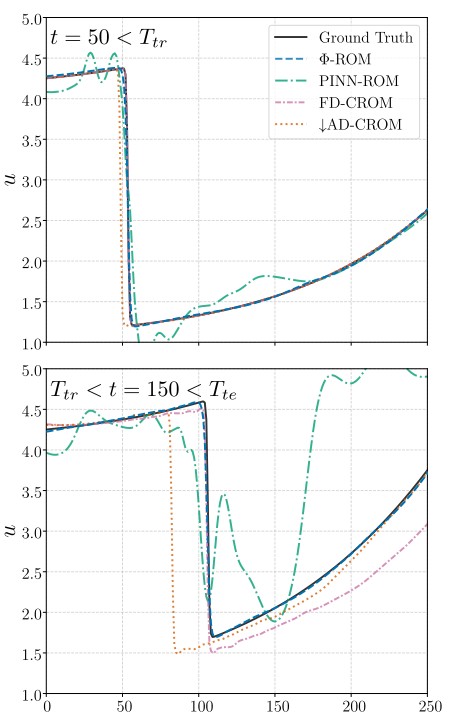

(a) Comparing the predicted solution to 1D Burgers' equation for an unseen initial state during (top) interpolation and (bottom) extrapolation periods.

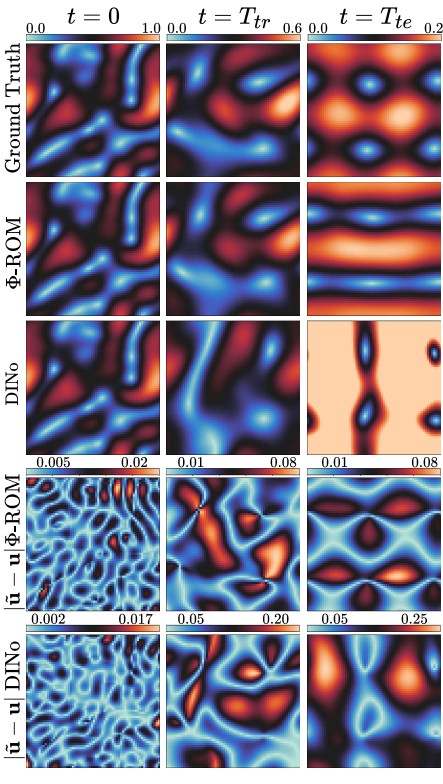

(b) Forecasted fields and their corresponding absolute errors for an unseen initial condition of N-S. Plotted are velocity magnitudes.

Figure 3: Forecasting Burgers' (left) and N-S (right) test samples.

## 4.2 Physics-informed strategies

We first compare Φ-ROM with two other alternative techniques for constructing physics-informed ROMs, namely CROM [21] and PINN-ROM (see Remark 3.2). For reference, we also provide comparisons with the data-driven method of DINo [13]. In this section, we only discuss relatively simple PDEs, namely 2D Diffusion (linear) and 1D Burgers' (nonlinear) equations. All models have the same decoder architecture, while Φ-ROM, PINN-ROM, and DINo also have the same dynamics network architecture. We evaluate CROM in three settings: (FD-CROM) where the temporal evolution is based on the full grid using finite difference discretization, (AD-CROM) where the spatial derivatives of the PDE are calculated using AD on the full grid, and (↓AD-CROM) where the spatial derivatives are calculated using AD but on a sub-sampled grid to achieve meaningful reduction. In all three settings, the training is identical, and the same trained decoder is used for evaluation.

Table 2: Forecasting errors (RNMSE) of Φ-ROM and DINo for N-S and KdV problems trained with $\mathbf{U}_{tr}$ on grid $\mathcal{X}_{tr}$. Errors are computed for initial conditions (ICs) sampled from $\mathbf{U}_{tr}$ or $\mathbf{U}_{te}$ and observed on test grid $\mathcal{X}_{te}$, and are averaged over either the interpolation window $[0, T_{tr}]$ or the extrapolation window $[T_{tr}, T_{te}]$.

| | | N-S | | | | KdV | | | |
| | | $[0, T_{tr}]$ | | $[T_{tr}, T_{te}]$ | | $[0, T_{tr}]$ | | $[T_{tr}, T_{te}]$ | |
| Time IC Set | | $\mathbf{U}_{tr}$ | $\mathbf{U}_{te}$ | $\mathbf{U}_{tr}$ | $\mathbf{U}_{te}$ | $\mathbf{U}_{tr}$ | $\mathbf{U}_{te}$ | $\mathbf{U}_{tr}$ | $\mathbf{U}_{te}$ |
|---|---|---|---|---|---|---|---|---|---|
| **Full training** | | | | $\mathcal{X}_{tr} = \mathcal{X}_{\mathcal{S}} = \mathcal{X}_{te}$ | | | | | |
| | Φ-ROM | 0.064 | **0.170** | **0.136** | **0.373** | 0.219 | **0.233** | 0.475 | **0.486** |
| | DINo | **0.036** | 0.580 | 0.692 | 1.543 | **0.098** | 0.459 | **0.399** | 0.728 |
| | | | | $\mathcal{X}_{tr} = \mathcal{X}_{\mathcal{S}}, \ |\mathcal{X}_{te}| = 5\%|\mathcal{X}_{\mathcal{S}}|$ | | | | | |
| | Φ-ROM | 0.066 | **0.170** | **0.139** | **0.378** | 0.219 | **0.233** | 0.475 | **0.487** |
| | DINo | **0.041** | 0.580 | 0.692 | 1.580 | **0.099** | 0.459 | **0.399** | 0.728 |
| | | | | $\mathcal{X}_{tr} = \mathcal{X}_{\mathcal{S}}, \ |\mathcal{X}_{te}| = 2\%|\mathcal{X}_{\mathcal{S}}|$ | | | | | |
| | Φ-ROM | **0.078** | **0.177** | **0.160** | **0.393** | 0.680 | 0.764 | 0.901 | 0.967 |
| | DINo | 0.087 | 0.587 | 0.693 | 1.558 | **0.523** | **0.639** | **0.677** | **0.796** |
| **Sparse training** | | | | $|\mathcal{X}_{tr}| = 5\%|\mathcal{X}_{\mathcal{S}}|, \ \mathcal{X}_{te} = \mathcal{X}_{\mathcal{S}}$ | | | | | |
| | Φ-ROM | 0.077 | **0.192** | **0.148** | **0.397** | 0.238 | **0.248** | 0.485 | **0.499** |
| | DINo | **0.046** | 0.584 | 0.717 | 1.450 | **0.165** | 0.543 | **0.498** | 0.851 |
| | | | | $|\mathcal{X}_{tr}| = 2\%|\mathcal{X}_{\mathcal{S}}|, \ \mathcal{X}_{te} = \mathcal{X}_{\mathcal{S}}$ | | | | | |
| | Φ-ROM | **0.092** | **0.189** | **0.169** | **0.394** | **0.268** | **0.280** | **0.546** | **0.567** |
| | DINo | 0.183 | 0.594 | 0.726 | 1.517 | 0.759 | 0.902 | 1.222 | 1.396 |

Table 1 shows the forecasting errors of all models on the Diffusion and Burgers' problems. For the linear Diffusion problem with smooth Gaussian blobs as the initial conditions, both PINN-ROM and Φ-ROM closely outperform DINo, and they all perform better than CROM, even without reducing the spatial grid. We note that AD-CROM slightly improves FD-CROM, showing its effectiveness in simple linear problems. However, for the non-linear Burgers', both AD-CROM and PINN-ROM fail grossly to capture the dynamics and exhibit higher errors than FD-CROM and DINo (see Fig. 3a). Comparing FD-CROM with Φ-ROM in Fig. 3a, both methods accurately reproduce the dynamics of the non-linear PDE (especially the travelling shock wave) during the training window, but unlike Φ-ROM, FD-CROM fails to forecast the solution evolution for $t > T_{tr}$. Failure of AD-CROM and PINN-ROM in the non-linear case highlights the challenges with optimizing PINN-type losses as studied in the literature (e.g. spectral bias) [29–31].

In contrast to Φ-ROM, CROM [21] evolves the solution directly in physical space using the exact form of the governing PDE. Nevertheless, its computational savings do not stem from a reduced representation of a nonlinear manifold but rather from using a limited number of sub-sampled grid points relative to the full-order discretization (i.e. ↓AD-CROM). Moreover, constructing PDE residuals is not always feasible when sub-sampling an INR that does not output all required physical fields. For instance, solving the Navier-Stokes equations requires both pressure and velocity fields, and thus cannot be done with an INR that only predicts velocity. Without sub-sampling, CROM offers no computational savings and still incurs a significant cost during online inference, as it requires solving a least-squares optimization problem at every time step in the full space to recover the corresponding reduced latent coordinates. For these reasons and considering the PINN-ROM inaccuracies highlighted above, we only compare Φ-ROM with DINo in the remainder of this paper.

### 4.3 Forecasting performance on full grid

**(N-S)** We trained Φ-ROM and DINo with identical networks on N-S with $M_{tr} = 256$ trajectories and $T_{tr} = 50$ time steps on a $64 \times 64$ regular grid $\mathcal{X}_{tr} = \mathcal{X}_{\mathcal{S}}$ and evaluated them with $M_{te} = 64$ trajectories of unseen initial conditions with $T_{te} = 200$. Table 2 (top rows) reports the average forecasting errors for both test and train initial conditions (see also Fig. 3b for a visual comparison).

While DINo achieves a low average error of 0.02 on the training initial conditions and time horizon, $\Phi$-ROM consistently outperforms DINo in test samples and temporal extrapolation. Notably, $\Phi$-ROM performs more than four times better than DINo in the test extrapolation, demonstrating its superior generalization and temporal stability. Both $\Phi$-ROM and DINo are capable of forecasting initial conditions on new (and irregular) spatial grids $\mathcal{X}_{te} \neq \mathcal{X}_{tr}$. As shown in Table 2, $\Phi$-ROM is able to recover the full solution field from a partial observation of the initial conditions with only 5% and 2% of the grid points.

We further trained both $\Phi$-ROM and DINo with 128 and 512 trajectories to evaluate their data efficiency. Fig. 4 shows how their forecasting errors (with initial conditions sampled from either test or training sets) change as the training dataset size increases. We observe that the physics-informed training of $\Phi$-ROM significantly improves the generalization and prevents overfitting compared to DINo, especially with fewer training trajectories. With $M_{tr} = 512$, $\Phi$-ROM closes the gap between the train and test errors.

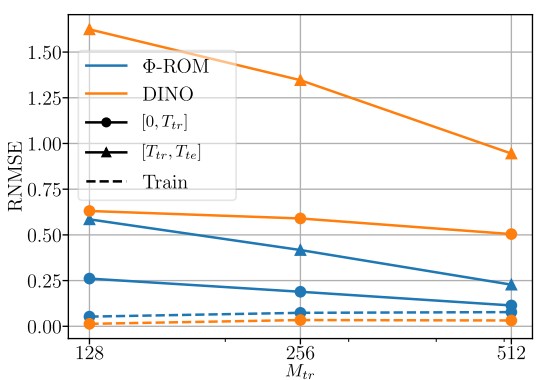

Figure 4: RNMSE for N-S based on $\Phi$-ROM and DINo for $[0, T_{tr}]$ and $[T_{tr}, T_{te}]$ with increasing size of the training dataset, showing data efficiency.

**(KdV)** Table 2 shows test and train forecast errors for the KdV equation. Both models are trained with $M_{tr} = 512$ trajectories of $T_{tr} = 40$ time steps on a $64 \times 64$ regular grid and evaluated with $M_{te} = 64$ test trajectories for $T_{te} = 80$ time steps. $\Phi$-ROM again outperforms DINo on test interpolation and extrapolation errors, while DINo fails to generalize to unseen initial conditions. Notably, however, both models perform poorly in sub-sampled inference with only 2% grid observation of the initial conditions.

**(LBM)** For the problem of flow over a cylinder, we keep initial conditions the same and evaluate the performance of $\Phi$-ROM for different PDE parameters (here $\beta$ represents the Reynolds number). We train the LBM models, with $\beta_{tr}$ ranging from 100 to 200 for training and $\beta_{te}$ ranging from 200 to 300 for parameter extrapolation. We use 40 training trajectories with $T_{tr} = 100$ and evaluate with $T_{te} = 125$ time steps. Table 3 reports forecasting errors for LBM with unseen parameters $\beta_{tr}$ and $\beta_{te}$. DINo captures the dynamics of the parameters within the training range and achieves smaller errors than $\Phi$-ROM. However, for out-of-distribution parameters $\beta_{te}$, DINo fails to forecast the PDE with the unseen dynamics, while $\Phi$-ROM maintains a small error. Both models are able to recover the solutions when only 2% of the training grid is used for inference.

Table 3: LBM forecast errors (RNMSE) for interpolation and extrapolation parameter (i.e., Reynolds number) and time intervals.

| Time
$\beta$ | $[0, T_{tr}]$ | | $[T_{tr}, T_{te}]$ | |
|---|---|---|---|---|
| | $\beta_{tr}$ | $\beta_{te}$ | $\beta_{tr}$ | $\beta_{te}$ |
| | $\mathcal{X}_{tr} = \mathcal{X}_{\mathcal{S}} = \mathcal{X}_{te}$ | | | |
| $\Phi$-ROM | 0.049 | **0.115** | 0.116 | **0.180** |
| DINo | **0.011** | 0.457 | **0.108** | 0.566 |
| | $\mathcal{X}_{tr} = \mathcal{X}_{\mathcal{S}},\ |\mathcal{X}_{te}| = 2\%|\mathcal{X}_{\mathcal{S}}|$ | | | |
| $\Phi$-ROM | 0.049 | **0.182** | 0.116 | **0.302** |
| DINo | **0.011** | 0.400 | **0.108** | 0.507 |
| | $|\mathcal{X}_{tr}| = 2\%|\mathcal{X}_{\mathcal{S}}|,\ \mathcal{X}_{te} = \mathcal{X}_{\mathcal{S}}$ | | | |
| $\Phi$-ROM | **0.065** | **0.188** | **0.150** | **0.303** |
| DINo | 0.369 | 0.412 | 0.420 | 0.439 |

## 4.4 Forecasting performance with sparse training

In the previous experiments, we trained all models with data on a regular grid, i.e. $\mathcal{X}_{tr} = \mathcal{X}_{\mathcal{S}}$. When trained with irregular and sparse observations at a small number of fixed locations, $\Phi$-ROM is still able to learn the dynamics and recover the full solution fields. Table 2 (bottom rows) reports the forecasting errors for sparse training of N-S and KdV with training grids that are 5% and 2% of their full grids in size. In all cases, $\Phi$-ROM maintains its accuracy close to the full training scenario, while errors for DINo grow significantly, especially in the 2% case. Similarly, we evaluate sparse training for LBM with 2% grid size (see bottom rows of Table 3). $\Phi$-ROM again recovers the full solution

fields and maintains its generalization to out-of-domain parameters, while DINo fails to capture the dynamics from the partial observations.

## 5   Conclusion

We introduced $\Phi$-ROM, a physics-informed ROM that embeds the governing physics in the reduced latent space by learning the dynamics directly from a differentiable PDE solver. We showed that compared to other physics-informed strategies and data-driven methods for constructing ROMs, $\Phi$-ROM generalizes better to unseen dynamics induced by new initial conditions or parameters, forecasts beyond the training temporal horizon, and recovers the full solution fields when trained or tested with partial observations. The model provides accelerated forecasting within a reduced space and benefits from a mesh-free construction that is continuous in space and time. Furthermore, $\Phi$-ROM adheres to the convergent discretization schemes of numerical PDE solvers when learning the temporal dynamics, making it robust to various physical phenomena compared to other physics-informed approaches.

Despite the promising results, we acknowledge that requiring access to differentiable PDE solvers limits $\Phi$-ROM's immediate applicability, although ongoing developments in differentiable programming offer promising avenues. The inclusion of the PDE solver in the training process also increases the training cost compared to purely data-driven methods. While we believe the superior data efficiency of $\Phi$-ROM justifies the increased training costs (as demonstrated in Fig. 4), tractable physics-informed training for large-scale 3D problems would require further study and optimization of the dynamics loss and its components. Additionally, although $\Phi$-ROM was successfully trained with sparse data, extending the framework to data assimilation settings with limited *sensor* measurements is a compelling area for further exploration. Finally, this study focused on PDEs with first-order time derivatives, and extending to higher-order time-dependent (e.g. wave equations) or time-independent PDEs is left for future research.

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

# Appendix

## A    Related works

**Implicit Neural Representations** Implicit Neural Representations (INRs) (also known as Neural Fields) are neural networks that map coordinates (such as spatiotemporal coordinates) to function or signal values and provide a continuous and differentiable representation of signals, complex geometries, and physical quantities [32–34]. Conditional INRs extend INRs by conditioning the represented function on additional parameters, such as a latent code, to model parametric functions [23] or their evolution in time [13, 21]. Using conditional INRs, Park et al. [23] proposed *auto-decoder* as an alternative to auto-encoders, where the discretization-dependent encoder network is replaced with an inversion step, which requires solving a least-squares problem to find the latent code, resulting in a continuous and discretization-free (or mesh-free) encoding and decoding. In this work, we similarly use conditional INRs for autodecoding to represent the PDE solution fields.

**Reduced order modeling** Traditionally, linear reduction methods (e.g. PCA) have been employed to identify reduced basis functions for constructing ROMs using a collection of snapshots [1, 2]. In projection-based reduced basis methods, the governing equations are then intrusively modified to arrive at a truncated set of parameterized equations in the reduced space, while non-intrusive ROMs rely on interpolation or regression between reduced basis coefficients for a new parameter or time value (see [3] for more details). For highly nonlinear systems where the Kolmogorov $n$-width decays slowly, linear subspace methods struggle to capture the underlying patterns of the system efficiently and accurately [4, 22]. A prominent method for constructing nonlinear ROMs, first proposed by Lee and Carlberg [4], leverages auto-encoders to learn a nonlinear, compact, and low-dimensional manifold of the solution space. Almost always, this reduced space is employed for inferring unseen dynamics at new parameter and time values (except for CROM [21] to be discussed below). In these cases, when the governing PDEs are parameterized or time-dependent, the realized latent space also needs to become parameterized or time-dependent. This has been achieved either by (i) discovering equations that govern the latent space through sparse symbolic regression [6, 7], for example,e using SINDy [5] or (ii) training a secondary network to incorporate the additional dependencies of the latent space. The latter includes a whole host of methods that vary in their choice of this secondary network to learn the dynamic evolution [8–15] or parametric dependency [25, 35]. Notably, Yin et al. [13] proposed DINo as a time and space continuous ROM that uses a conditional INR to reduce the spatial dimension and a Neural ODE [36] to evolve the latent coordinates.

In contrast to the aforementioned works, CROM [21] evolves the solution directly in the physical space using the exact form of the continuous governing PDE (similar to PINNs) at the inference time. Although CROM enforces the underlying equations, making it more interpretable than purely data-driven approaches, its computational savings do not stem from a reduced latent representation of a nonlinear manifold. Instead, it relies on the use of a limited number of sub-sampled grid points that are reconstructed and used to calculate the temporal derivatives of the PDE using automatic differentiation. The time derivatives are then used to find the solution at the next time step which is subsequently encoded to its corresponding latent code. However, due to the errors in the gradients when taking the subsampled time derivatives, CROM is not always able to perform subsampling and instead uses the discretized (e.g. finite-differences) PDE on the full grid to compute the time derivatives. As such, CROM essentially solves the full-order PDE in such scenarios.

Our proposed framework of $\Phi$-ROM adopts a similar setup to DINo [13], but benefits from a physics-informed training strategy. $\Phi$-ROM directly trains the dynamics network with time derivatives of the reconstructed fields, obtained from a differentiable solver, in contrast to DINo, which relies on time-integration during training using neural ODEs to evolve the latent coordinates. We also leverage hyperreduction [22] for fast and efficient training of the dynamics network in $\Phi$-ROM, unlike other similar approaches that also rely on the time derivatives of the reconstructed fields [12, 19]. Moreover, $\Phi$-ROM differs substantially from CROM in the sense that it enforces the governing physics in the latent space and evolves the solution directly in the latent space during the training as opposed to the full physical space.

**Machine learning for physics** Apart from ROMs, Neural Operators [37] and Physics-Informed Neural Networks (PINNs) [32] are the other prominent machine learning methods for solving PDEs. Neural Operators learn mappings between function spaces and can solve parameterized PDEs at either fixed times [37, 38] or autoregressively [39, 40]. Fourier Neural Operators perform a

kernel convolution with the Fourier basis to solve PDEs on regular grids [38], while graph neural operators extend to irregular grids and free-form geometries [39, 37], although they are still limited in generalizing to unseen grids. While Neural Operators are often completely data-driven methods, given a PDE and its initial/boundary conditions, PINNs use AD and INRs for solving PDEs without requiring labelled data. While effective for solving simple PDEs, the original formulation of PINNs introduces optimization challenges with stiff equations due to its complex loss landscape [41, 29, 31]. Furthermore, while continuous in time and space, PINNs are limited to a single parameterization of a PDE and are unable to extrapolate beyond their training time horizon. To alleviate these limitations, similar physics-informed training methods have been proposed for training neural operators [42].

**Differentiable solvers** Although numerical solvers have long been studied and developed in computational physics and engineering, the growth of machine learning and the development of differentiable programming have prompted the development of differentiable solvers that can be intertwined with machine learning pipelines. JAX-CFD [27], XLB [28], and JAX-Fluids [43] for computational fluid dynamics, Torch Harmonics [44] for spherical signal processing, j-Wave [45] for acoustic simulation, and JAX-FEM [46] for finite element methods are some recent examples of such solvers implemented in JAX and PyTorch. Pioneered by Um et al. [47], neural networks have been used adjacent with numerical solvers to improve their accuracy and convergence by learning corrections to the numerical errors caused by discretization [27]. More recently, Apebench [26] proposed a training algorithm for auto-regressive neural PDE surrogates where a differentiable solver makes time-steps on the network predictions to improve the forecasting stability. In both methods, the interaction between the neural network and the solver is crucial for the neural network to learn the underlying physics of the system. $\Phi$-ROM is more closely related to the latter, where the neural network models the physics, and the solver is only used for training. However, $\Phi$-ROM creates the correspondence between the solver and the neural network in the latent space of a ROM rather than the full physical space. To demonstrate $\Phi$-ROM in this paper, we will report results based on JAX-CFD, XLB, and Apebench libraries, where each library is used for a different PDE.

# B    Training details

## B.1    Training procedure

We train $\Phi$-ROM by minimizing the reconstruction and dynamics losses together in two phases. In the first warm-up phase, $\Gamma$ and $D_\theta$ are trained with only the reconstruction loss, and $\Psi_\phi$ is trained with the dynamics loss. The warm-up epochs ensure that the reconstruction loss decreases to a small value before the full training phase begins. In the full training phase, the aggregate loss in Eq. (10) is minimized w.r.t. all model parameters. See Algorithm 1 for the detailed training procedure. We choose AdamW [48] optimizer to train $\Phi$-ROM and apply exponential learning rate decay with a decay rate of $0.985$ every 50 epochs. (See Appendix F.3 for complete hyperparameters.) Note that, unlike Yin et al. [13], both loss terms are minimized together using the same optimizer in both training phases.

## B.2    Dynamics loss and hyper-reduction

We used stochastic hyper-reduction to randomly sample a subset of the points in the time derivatives given by the solver and solve the least-squares problem for the dynamics loss with only the selected points to reduce the training computational cost. See Algorithm 2 for a step-by-step description of the dynamics loss. Note that the random reduced coordinates $\mathcal{X}_\mathcal{S}^\downarrow$ are independently sampled for each training snapshot at every training epoch.

We solve the least squares problem on line 7 in Algorithm 2 by first computing the QR decomposition of the reduced Jacobian matrix $J^\downarrow \in \mathbb{R}^{\gamma Nm \times k}$:

$$J^\downarrow = QR,$$

where $Q \in \mathbb{R}^{\gamma Nm \times \gamma Nm}$ and $R \in \mathbb{R}^{\gamma Nm \times k}$ are orthogonal and upper triangular matrices, respectively. Replacing $J^\downarrow$ with $QR$ in the least squares problem and multiplying both sides by $Q^\top$ we get:

$$J^\downarrow \dot{\alpha}^* = \dot{\mathbf{u}}^\downarrow \xrightarrow{J^\downarrow = QR} QR\dot{\alpha}^* = \dot{\mathbf{u}}^\downarrow \xrightarrow{\times Q^\top} R\dot{\alpha}^* = Q^\top \dot{\mathbf{u}}^\downarrow.$$

---

**Algorithm 1** Training procedure of Φ-ROM

---

1: **Input:** $\mathbf{U}_{tr} = \left\{ \mathbf{u}_i^{0:T_{tr}}, \beta_i, \mathcal{X}_{tr} \mid i \in \{0, \ldots, M_{tr}\} \right\}$
2: $\Gamma \leftarrow \{\alpha_i^t = \mathbf{0} \mid i \in \{0, \ldots, M_{tr}\}, t \in \{0, \ldots, T_{tr}\}\}$
3: Randomly initialize $\theta$ and $\phi$
4: **for** each warm-up epoch **do**
5:      **for** each $(\mathbf{u}_i^t, \alpha_i^t) \in (\mathbf{U}_{tr}, \Gamma)$ **do**
6:          $\theta^*, \Gamma^* \leftarrow \lambda \nabla_{\theta, \Gamma} L_{rec}(\mathbf{u}_i^t; \theta, \alpha_i^t)$
7:          $\phi^* \leftarrow (1 - \lambda) \nabla_{\phi} L_{dyn}(\theta, \alpha_i^t, \phi)$
8:          $\theta, \Gamma, \phi \leftarrow optimizer(\theta^*, \Gamma^*, \phi^*)$
9:      **end for**
10: **end for**
11: **for** each training epoch **do**
12:      **for** each $(\mathbf{u}_i^t, \alpha_i^t) \in (\mathbf{U}_{tr}, \Gamma)$ **do**
13:          $\theta^*, \Gamma^*, \phi^* \leftarrow \nabla_{\theta, \Gamma, \phi} L_{\Phi\text{-ROM}}(\mathbf{u}_i^t; \theta, \alpha_i^t, \phi)$
14:          $\theta, \Gamma, \phi \leftarrow optimizer(\theta^*, \Gamma^*, \phi^*)$
15:      **end for**
16: **end for**

---

Since $R$ is an upper-triangular matrix, we can solve the problem for $\dot{\alpha}^*$. In practice, we used NumPy's QR decomposition and SciPy's "solve_triangular" implementations in JAX for differentiability.

Moreover, we use forward-mode AD to compute the Jacobian matrix required in solving Eq. (7). This is because $D_\theta : \mathbb{R}^k \mapsto \mathbb{R}^{\gamma N \times m}$ and $k \ll \gamma N m$. Note that forward mode AD achieves $\mathcal{O}(k)$ memory complexity and $\mathcal{O}(k T_D)$ time complexity (where $T_D$ is the computational cost of the forward pass), which is significantly cheaper than the $\mathcal{O}(\gamma N m)$ memory and $\mathcal{O}(\gamma N m T_D)$ time complexity of the standard reverse-mode AD.

---

**Algorithm 2** Dynamics loss of Φ-ROM

---

1: **function** $L_{dyn}(\theta, \phi, \alpha)$
2:      $\hat{\mathbf{u}} \leftarrow D_\theta(\alpha, \mathcal{X}_\mathcal{S})$         $\triangleright$ Reconstruct the field on the solver grid ($\hat{\mathbf{u}} \in \mathbb{R}^{N \times m}, N = |X_\mathcal{S}|$)
3:      $\dot{\hat{\mathbf{u}}} \leftarrow (\hat{\mathbf{u}}, \mathcal{S})$         $\triangleright$ Compute time derivative, see Appendix B.4
4:      $\mathcal{X}_\mathcal{S}^{\downarrow} \sim \mathcal{X}_\mathcal{S}$         $\triangleright$ Randomly sample $\gamma N$ coordinates (where $N = |\mathcal{X}_\mathcal{S}|$)
5:      $\dot{\hat{\mathbf{u}}}^{\downarrow} \leftarrow \dot{\hat{\mathbf{u}}}|_{\mathcal{X}_\mathcal{S}^{\downarrow}}$         $\triangleright$ Sample time derivatives at $\mathcal{X}_\mathcal{S}^{\downarrow}$ and flatten ($\dot{\hat{\mathbf{u}}}^{\downarrow} \in \mathbb{R}^{\gamma N m}$)
6:      $J^{\downarrow} \leftarrow \nabla_\alpha D_\theta(\alpha, \mathcal{X}_\mathcal{S}^{\downarrow})$         $\triangleright$ Jacobian via forward mode AD ($J^{\downarrow} \in \mathbb{R}^{\gamma N m \times k}$)
7:      $\dot{\alpha}^* \leftarrow \mathrm{argmin}_{\dot{\alpha}} \left\| \dot{\hat{\mathbf{u}}}^{\downarrow} - J^{\downarrow} \dot{\alpha} \right\|$         $\triangleright$ Solve least-squares ($\dot{\alpha}^* \in \mathbb{R}^k$)
8:      **return** $\ell(\Psi_\phi(\alpha), \dot{\alpha}^*)$         $\triangleright$ Return dynamics loss
9: **end function**

---

### B.3   Loss measure

We use RNMSE to measure the training error in both the reconstruction and the dynamics losses, ensuring that both loss terms are of the same scale. For a reconstructed field $\hat{\mathbf{u}}$ and a label $\mathbf{u}$ on the same grid $\mathcal{X}$, the error is calculated as

$$\ell(\hat{\mathbf{u}}, \mathbf{u}) = \frac{1}{m} \sum_{i=1}^{m} \frac{\|_i\hat{\mathbf{u}} - {}_i\mathbf{u}\|_\mathcal{X}}{\|_i\mathbf{u}\|_\mathcal{X}}, \tag{12}$$

where $\| \cdot \|_\mathcal{X}$ is the Frobenius norm over the spatial axes, and the average is taken over the field channel axis (e.g. velocity directions) indexed by $i$.

The RNMSE for two latent time derivative $\dot{\alpha}_1, \dot{\alpha}_2 \in \mathbb{R}^k$ in the dynamics loss is calculated slightly differently as

$$\ell(\dot{\alpha}_1, \dot{\alpha}_2) = \frac{\|\dot{\alpha}_1 - \dot{\alpha}_2\|_2}{\|\mathrm{ST}(\dot{\alpha}_2)\|_2}, \tag{13}$$

where $\| \cdot \|_2$ is the 2-norm and ST stops the gradients from backpropagating through the denominator. Since $\dot{\alpha}_2$ is a function of trainable model parameters $\theta$ and $\alpha_2 \in \Gamma$, gradient stopping prevents the denominator and the gradients from growing large and stabilizes the training.

### B.4 Computing the time derivatives

We used numerical PDE solvers to compute $d\hat{\mathbf{u}}/dt$ or the time derivatives of a reconstructed state $\hat{\mathbf{u}}$. If $d\mathbf{u}/dt$ is not an output of a differentiable solver after one time step, we may resort to a first-order approximation of the time derivative to compute $d\mathbf{u}/dt$ given by:

$$\dot{\hat{\mathbf{u}}} = \frac{\mathcal{S}[\hat{\mathbf{u}}] - \hat{\mathbf{u}}}{\Delta t_\mathcal{S}},$$

where $\Delta t_\mathcal{S}$ is the step size of the solver. This approach proved to be sufficient in our N-S, KdV, and LBM experiments.

Some numerical PDE solvers, on the other hand, may be sensitive to noise and errors in the reconstructed fields and return corrupted time derivatives that are not suitable for training. In such cases, we let the solver apply multiple time steps on the reconstructed field to correct the noise and calculate an averaged time derivative. We employed this method in our LBM experiments by applying five consecutive solver steps.

## C  Inference details

Inference in $\Phi$-ROM consists of three steps (similar to DINo): **(1)** inversion, **(2)** integration, and **(3)** decoding. Algorithm 3 provides the pseudo-code for these three steps. **(1)** Given an initial condition $\mathbf{u}^{t_0}$ discretized on a (possibly irregular) grid $\mathcal{X}$, $\Phi$-ROM first solves the least squares problem defined in Eq. (4) to find the corresponding latent coordinates $\hat{\alpha}^0$. The inversion is performed by minimizing the same RNMSE loss defined in Eq. (12) using AdamW optimizer (same as the training optimizer) with a zero initialization for $\hat{\alpha}^0$. We perform 1000 optimization steps across all our experiments with a learning rate of 0.1. **(2)** For a desired time $T_* > t_0$, the dynamics network is integrated in time to find the final latent coordinate $\tilde{\alpha}^{T_*}$. In this work, we used a third-order explicit Runge-Kutta ODE solver with adaptive step size, Bogacki–Shampine method [49], for numerical integration. We did not observe any improvements by using other higher-order ODE solvers. **(3)** The predicted solution at time $T_*$ is obtained by simply decoding $\tilde{\alpha}^{T_*}$ on any grid $\mathcal{X}'$, where $\mathcal{X}$ and $\mathcal{X}'$ can be different. Note that the model may not have seen the integration time $T_*$ during training, and the model is continuous in time.

---

**Algorithm 3** Inference with $\Phi$-ROM

---

1: **Input:** $(\mathbf{u}^{t_0}, \mathcal{X}), \mathcal{X}', T_*, D_\theta, \Psi_\phi$
2: $\hat{\alpha}^{t_0} \leftarrow \mathbf{0}$
3: **for** each inversion step **do**                          $\triangleright$ Inversion $(D_\theta^\dagger(\mathbf{u}^{t_0}, \mathcal{X}))$
4:     $\quad \alpha^* \leftarrow \nabla_\alpha \ell(D_\theta(\hat{\alpha}^{t_0}, \mathcal{X}), \mathbf{u}^{t_0})$
5:     $\quad \hat{\alpha}^{t_0} \leftarrow optimizer(\alpha^*)$
6: **end for**
7: $\tilde{\alpha}^{T_*} \leftarrow ODESolve(\hat{\alpha}^{t_0}, t_0, T_*)$              $\triangleright$ Integrating the latent coordinates
8: $\tilde{\mathbf{u}}^{T_*} \leftarrow D_\theta(\tilde{\alpha}^{T_*}, \mathcal{X}')$              $\triangleright$ Decoding the final state

---

## D  Detailed description of the datasets

Table 4 summarizes the datasets used in the experiments. The datasets are generated using different PDEs and solvers, and the details of each dataset are described below. In all experiments, the same solver used for data generation is also used for training $\Phi$-ROM.

Table 4: Summary of the datasets used in the experiments. $M_{tr}, M_{te}, M_{val}$ are the number of training, test, and validation trajectories, $T_{tr}$ and $T_{te}$ are the number of training and testing time steps, and $m$ is the number of output scalar fields.

| | $M_{tr}$ | $M_{te}$ | $M_{val}$ | $T_{tr}$ | $T_{te}$ | $\mathcal{X}_S$ | $m$ |
|---|---|---|---|---|---|---|---|
| Diffusion | 100 | 32 | 16 | 25 | 200 | $42 \times 42$ | 1 |
| Burgers' | 8 | 9 | – | 100 | 200 | 256 | 1 |
| N-S | 256 | 64 | 16 | 50 | 200 | $64 \times 64$ | 2 |
| KdV | 512 | 64 | 16 | 40 | 80 | $64 \times 64$ | 1 |
| LBM | 40 | $40 + 40$ | 4 | 100 | 125 | $180 \times 82$ | 2 |

## D.1 Diffusion equation

We consider the 2D diffusion equation with constant diffusivity $\kappa = 2$, defined on a bounded spatial domain with zero Dirichlet boundary conditions.

$$\frac{\partial u}{\partial t} = \kappa \left( \frac{\partial^2 u}{\partial x^2} + \frac{\partial^2 u}{\partial y^2} \right), \quad \kappa = 2,$$

$$u(x, y, t) = 0 \quad \text{for } (x, y) \in \partial \Omega.$$

We let $\Omega = [-20, 20]^2$ and sample Gaussian blobs with means $\mu_x, \mu_y \in [-12, 12]$, standard deviations ranging from 3 to 10, and amplitudes ranging from 0.5 to 2 sampled uniformly for the initial conditions. We use a uniform grid of $42 \times 42$ points for the spatial domain and a time step of 0.1 for the simulation and solve the equation using a finite-difference solver. We generate 100 training and 32 test trajectories with $T_{tr} = 25$ and $T_{te} = 200$.

## D.2 Burgers' equation

We consider a 1D inviscid Burgers' equation,

$$\frac{\partial w}{\partial t} + 0.5 \frac{\partial w^2}{\partial x} = 0.02 e^{\mu x},$$

parameterized by $\mu$ in the source term. We use the same initial profile as in Chen et al. [21] and Lee and Carlberg [4]. Training parameters ($\mu$) are sampled from $[0.015, 0.03]$ to generate $M_{tr} = 8$ training trajectories with

$$\mu \in \{0.015, 0.0171, 0.193, 0.0214, 0.0236, 0.0257, 0.0279, 0.03\}.$$

We similarly generate $M_{te} = 9$ test trajectories with

$$\mu \in \{0.0129, 0.0161, 0.0182, 0.0204, 0.0225, 0.0246, 0.0268, 0.0289, 0.0321\},$$

which includes two extrapolating parameters. Trajectories are simulated for 200 time steps with a finite-difference solver on a 256 1D grid, where the first 100 time steps are used for training and the rest for extrapolation.

## D.3 Navier-Stokes decaying turbulence

We model a decaying turbulence problem defined by the incompressible Navier-Stokes equations on a periodic domain in 2D as

$$\frac{\partial u}{\partial t} = -\nabla \cdot (u \otimes u) + \frac{1}{Re} \nabla^2 u - \frac{1}{\rho} \nabla p, \qquad \nabla \cdot u = 0,$$

where $u$ is the velocity field, $p$ is the pressure, $Re = UL/\nu$ is the Reynolds number and $\rho$ is the density. The spatial domain is a periodic box of size $L = \pi$ discretized on a uniform grid of $64 \times 64$ points. The initial conditions are generated randomly and filtered to be divergence-free and have a maximum velocity magnitude of $U = 1$ and a peak wavenumber of 3. Density is set to 1 and the kinematic viscosity is $\nu = 0.01$, giving $Re \approx 310$.

We used JAX-CFD, a JAX-based library that provides finite volume solvers for differentiable computational fluid dynamics simulations [27], to simulate the Navier-Stokes equations. We chose a time step size of $\Delta t_S = 0.0245$, and saved the velocity field every 5 steps for a total of $T = 200$ (1000 solver steps). The dataset contains a total of 1024 trajectories from random initial conditions, where $M_{tr} = [128, 256, 512]$, $M_{te} = 64$, and 32 trajectories are used for validation. Training is performed on the first $T_{tr} = 50$ time steps of the trajectories, and testing is performed on $T_{te} = 200$ time steps.

### D.3.1 Dataset for training with sparse irregular grid

We prepare the training data for experiments in Section 4.4 by first upscaling the $64 \times 64$ solver grid to a $128 \times 128$ grid by linear interpolation, and then randomly sampling 5% and 2% of the original grid size, i.e., $\lfloor 0.05 \times 64^2 \rfloor = 204$ and $\lfloor 0.02 \times 64^2 \rfloor = 81$ probing locations are sampled from the upscaled $128 \times 128$ grid for the 5% and 2% cases, respectively. We perform the linear interpolation and upscaling of the grid to ensure that the training grid is not a subset of the solver grid, which is used for inference. The selected sparse locations are fixed and the same across all training snapshots and trajectories.

### D.4 Korteweg–De Vries equation

Exponax library provides spectral solvers for semi-linear PDEs [26]. We use Exponax to simulate the KdV equation defined as:

$$\frac{\partial u}{\partial t} - 3\nabla \cdot (u \otimes u) + \mathbf{1} \cdot (\nabla \odot \nabla \odot (\nabla u)) = 0.03 \left( (\nabla \odot \nabla) \cdot (\nabla \odot \nabla) \right) u$$

in a 2D periodic domain. Initial conditions are randomly sampled from a random Fourier series with a maximum wavenumber of 2 and normalized to have a maximum amplitude of 1. The periodic domain length is 15 and is discretized on a uniform grid of $64 \times 64$ points. The training dataset contains $M_{tr} = 512$ trajectories with $T_{tr} = 40$ with $\Delta t_S = 0.04$, and the test dataset contains $M_{te} = 64$ trajectories with $T_{te} = 80$, and 16 trajectories are used for validation.

### D.4.1 Dataset for training with sparse irregular grid

We sampled the sparse data for KdV similar to N-S as explained in Appendix D.3.1.

### D.5 Lattice Boltzmann equation

XLB is a differentiable library for solving computational fluid problems based on the Lattice Boltzmann Method (LBM) [28] and is implemented in JAX and Warp. The continuous Boltzmann equation for a distribution function $f(x, \xi, t)$ is defined as

$$\frac{\partial f}{\partial t} + \xi_i \frac{\partial f}{\partial x_i} + \frac{F_i}{\rho} \frac{\partial f}{\partial \xi_i} = \Omega(f), \tag{14}$$

where $\xi_i = dx_i/dt$ is the particle velocity in direction $i$, $F_i/\rho = d\xi_i/dt$ is the specific body force, and the source $\Omega(f)$ is the collision operator that represents the local redistribution of $f$ due to particle collisions. XLB solves the discretized version of this equation.

We used XLB in 2D to simulate flow around a circular cylinder with the D2Q9 lattice, BGK collision operator, and suitable boundary conditions representing the flow inlet and outlet, and the no-slip boundaries for a set of Reynolds numbers, $Re = UD/\nu$ where $U$ is the mean speed of the incoming flow, $D$ is the diameter of the cylinder, and $\nu$ is the kinematic viscosity. For sufficiently large Reynolds numbers, this configuration leads to the formation of the Kármán vortex street caused by vortex shedding. We generate the data on a $180 \times 82$ grid with Reynolds numbers ranging from 100 to 200 and simulate for 4000 solver steps. We discard the data from the first 3000 steps of the solver when the flow is still forming. Furthermore, since LBM makes very small steps, we save the remaining simulated data only every 5 steps. As such, each trajectory has 200 time steps, where $T_{tr} = 100$ are used for training and the rest are used for testing. However, we report the time extrapolation errors in the paper for only $T_{te} = 125$ as both $\Phi$-ROM and DINo fail to capture the long-horizon dynamics of LBM. (See Appendix H.3.). The training dataset consists of $N_{tr} = 40$ trajectories with linearly spaced Reynolds numbers $Re \in \beta_{tr} = [100, 200]$, and the test dataset contains $N_{te} = 80$ trajectories where 40 trajectories have $Re \in \beta_{tr}$ (parameter interpolation) and another 40 trajectories

have $Re \in \beta_{te} = [200, 300]$ (parameter extrapolation). In the paper, we report the test errors for parameter interpolation and parameter extrapolation settings separately.

Finally, the data generated by XLB is represented at mesoscopic scales on a D2Q9 lattice with discrete lattice velocities $c_i$ for $i \in \{1, \cdots, 9\}$. As we are interested in simulating the velocity, we convert the simulated distributions $f$ to velocity $u$ given by

$$u(x,t) = \frac{1}{\rho(x,t)} \sum_{i=1}^{9} c_i \, f_i(x,t), \tag{15}$$

$$\rho(x,t) = \sum_{i=1}^{i=9} f_i(x,t), \tag{16}$$

where $i$ indexes the discrete lattice directions, and $\rho$ is the density. See Ataei and Salehipour [28] for a detailed description of the LBM equations. We use the converted velocity fields for training DINo and use the distribution function $f$ for training $\Phi$-ROM as explained in Appendix D.5.2.

### D.5.1  Dataset for training with sparse irregular grid

We sample the sparse data for LBM similar to previous cases, except that the coordinates lying inside the cylinder are added to the sampled data to ensure that the cylinder (as a boundary condition) is fully represented in the data.

### D.5.2  Training and inference for LBM

**Training.** As noted earlier in Appendix D.5, XLB models the flow using the LBM equations and hence it represents the macroscopic variables (i.e. velocity and pressure) at mesoscopic scales. As such, the state variable $f(x,t)$ is a distribution function discretized on the D2Q9 lattice, i.e. $f(x,t) \in \mathbb{R}^9$. To train $\Phi$-ROM, we need to reconstruct the distribution functions $f$, instead of the velocity $u$, in order to pass the state to the solver and get the time derivatives. Thus, we train $\Phi$-ROM using the simulated distribution functions, where each snapshot has nine channels. Furthermore, we modify the reconstruction loss to further constrain the reconstructions using the velocity formula in Eq. (15). For a training snapshot $\mathbf{f} \in \mathbb{R}^{9 \times 180 \times 82}$ and the corresponding latent coordinate $\alpha$, the reconstruction loss for LBM is

$$L_{rec}(\mathbf{f}; \theta, \alpha) = \ell(\hat{\mathbf{f}}, \mathbf{f}) + \ell(\text{vel}[\hat{\mathbf{f}}], \text{vel}[\mathbf{f}]) + \ell(\text{rho}[\hat{\mathbf{f}}], \text{rho}[\mathbf{f}]), \tag{17}$$

where $\hat{\mathbf{f}} = D_\theta(\alpha, \mathcal{X})$ and rho$[\cdot]$ and vel$[\cdot]$ return the velocity and density as defined in Eq. 15. This reconstruction loss ensures that the reconstructed states meet the constraints required by the LBM solver, which would otherwise be sensitive to the errors in the reconstruction.

**Inference.** While we train with the distribution function, we still evaluate the model using the velocity fields (similar to DINo). As such, for an initial velocity field $\mathbf{u}$, we modify the reconstruction loss used for inversion in Eq. (4) to be

$$L_{rec}(\mathbf{u}; \alpha, \theta) = \ell(\text{vel}[D_\theta(\alpha)], \mathbf{u}). \tag{18}$$

Note that the above reconstruction loss for inference is different from Eq. 17, which was used during training. Using the new reconstruction loss, $\Phi$-ROM still operates on velocities. Furthermore, during evaluation, we calculate $\Phi$-ROM's errors by first converting its reconstructions to velocities and then measuring the errors, so that it can be compared to the results from DINo.

## E  Architecture details

$\Phi$-ROM consists of two neural network components: the INR decoder $D_\theta$ and the dynamics network $\Psi_\phi$. In this paper, we used the hyper decoder of [13] and a conditional SIREN [24] as decoders, depending on the problem, and an MLP with a parameterized Swish activation function (similar to Yin et al. [13]) as the dynamics network. We briefly describe the architectures below.

### E.1  Hyper decoder

Hyper decoder introduced by Yin et al. [13] is a conditional INR decoder where the spatial coordinates $x$ are the inputs to the network and the latent code $\alpha$ serves as "amplitude modulation" by setting

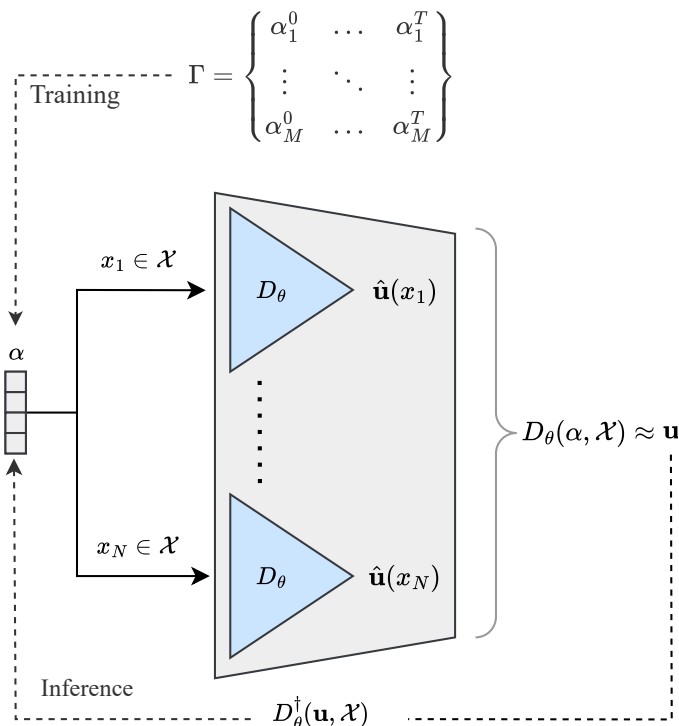

Figure 5: $D_\theta$ reconstructs the solution field one coordinate at a time and stacks the solution at all coordinates for the full field.

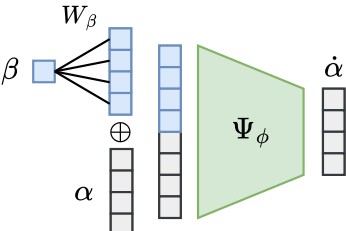

Figure 6: The parameterized dynamics network applies a linear transformation to the parameter $\beta$ before concatenating it with latent coordinates $\alpha$

the biases of the network. For a coordinate $x$ and a latent code $\alpha$, a hyper decoder with $L$ layers is defined as

$$z^0(x) = x,$$
$$z^l(x) = (W^{(l)}z^{(l-1)}(x) + b^{(l)} + \mu^{(l)}(\alpha)) \odot \left[\cos(\omega^{(l)}x), \sin(\omega^{(l)}x)\right] \quad \forall l \in [1, \ldots, L-1]$$
$$z^L(x) = (W^{(L)}z^{(L-1)}(x) + b^{(L)} + \mu^{(L)}(\alpha))$$

where $\mu^{(l)}(\alpha) = W'^{(l)}\alpha$ and $\theta = \{W^{(1\ldots L)}, b^{(1\ldots L)}, W'^{(1\ldots L)}, \omega^{(1\ldots L-1)}\}$ are the parameters of the network. Note that unlike Yin et al. [13], we include $\omega^{(l)}$ in the trainable parameters.

### E.2 SIREN decoder

SIREN [24] is an INR MLP with sine activation functions that is shown to be effective in learning continuous functions and solutions to PDEs [21, 41]. In this work, similar to Chen et al. [21], we also use a SIREN conditioned on the latent code $\alpha$, where the latent code is concatenated with the

input coordinates $x$ before being fed into the network (see Fig. 5). For a detailed description of the SIREN architecture, we refer the reader to Sitzmann et al. [24].

### E.3 Dynamics network

We adopt the MLP network with the parameterized Swish activation function used by DINo and extend it to parameterized PDEs by concatenating a linear transformation of the PDE parameters with the input latent coordinates (see Fig. 6). For a latent coordinate $\alpha \in \mathbb{R}^k$ and a PDE parameter $\beta \in \mathbb{R}^p$, the dynamics network $\Psi_\phi$ is defined as:

$$\Psi_\phi(\alpha, \beta) = \text{MLP}(\alpha \oplus \beta^*)$$
$$\beta^* = W_\beta \beta + b_\beta,$$

where $W_\beta \in \mathbb{R}^{k \times p}$ and $b_\beta \in \mathbb{R}^k$ along with the MLP parameters are the trainable parameters $\phi$, and $\oplus$ is concatenation. When the PDE is not parameterized, $\alpha$ is the only input to the network. The parameterized Swish activation function used in [13] is defined as:

$$\text{Swish}(x) = x \cdot \text{Sigmoid}(x \cdot \text{Softplus}(\omega)),$$

where $\omega$ is a trainable parameter. The output layer of the network is a linear layer of size $k$.

## F  Experimental settings

### F.1  Implementation

JAX enables fast and flexible differentiable programming by extending NumPy and SciPy and providing composable function transformations. We implemented $\Phi$-ROM in the JAX ecosystem using Equinox for building neural networks [50], Diffrax for numerical integration [51], and Optax for gradient optimization. As a result, any numerical solver implemented in JAX can be easily used for training $\Phi$-ROM, and the whole training pipeline, along with the solver step, is compiled and run together. In this work, we experimented with JAX-CFD [27], XLB [28], and Exponax [26] as PDE solvers, while other JAX-based solvers can be adopted as well. Furthermore, JAX transformations allow the training pipeline of $\Phi$-ROM to support multiple GPUs regardless of the implementation of the PDE solver.

As pointed out in Appendix B.2, we used the JAX implementation of SciPy and NumPy linear algebra utilities for differentiable integration with the training pipeline, support of multi-GPU execution, and easy batch operations. Moreover, as noted in Appendix B.2, the Jacobian matrix is computed using forward-mode AD, which is accessible through JAX's "jacfwd" function transformation. Finally, the *optimizer* and *ODESolve* function calls in Algorithms 1 and 3 are provided by Optax and Diffrax libraries implemented in JAX.

### F.2  Compute resources for training

We trained our Burgers' and Diffusion models on a single 48GB A6000 GPU, and KdV, N-S, and LBM models, each on four 40GB A100 GPUs. Training wall-clock times are as follows: 20 minutes for Burgers', 5 hours for Diffusion, 38 hours for N-S, 25 hours for KdV, and 18 hours for LBM.

### F.3  Hyperparameters

Tables 5 and 6 outline the network and training configurations, respectively. The batch sizes were chosen to be the largest that fit the available memory. In all experiments, we used the same decoder configuration for $\Phi$-ROM, DINo, PINN-ROM, and CROM, as well as the same dynamics network (or Neural ODE) for $\Phi$-ROM, DINo, and PINN-ROM.

We observed that a SIREN MLP as a decoder requires a much smaller latent space compared to the hyper decoder architecture from DINo. Furthermore, although SIREN is a periodic function with respect to the spatial coordinates, similar to the hyper decoder, it achieved better reconstruction and forecasting results in the Burgers' and LBM problems, which are not periodic. Furthermore, $\lambda = 0.5$ worked well in all our experiments except for LBM, where the solver appeared to be more sensitive to errors in the reconstructed fields.

Table 5: Network configuration for each problem. See Section E for a detailed description of the architectures.

| | Latent | $D$ | | | $\Psi$ | | |
|---|---|---|---|---|---|---|---|
| | $k$ | Arch. | Layers | Width | Arch. | Layers | Width |
| Diffusion | 16 | Hyper | 3 | 64 | MLP | 3 | 64 |
| Burgers' | 4 | SIREN | 4 | 32 | Param. MLP | 4 | 64 |
| N-S | 100 | Hyper | 3 | 80 | MLP | 4 | 512 |
| KdV | 100 | Hyper | 3 | 80 | MLP | 4 | 256 |
| LBM | 4 | SIREN | 6 | 64 | Param. MLP | 3 | 64 |

Table 6: Training hyperparameters for each problem.

| | Learning rate | Decay rate | Epochs | Warm-up epochs | $\lambda$ | $\gamma$ |
|---|---|---|---|---|---|---|
| Diffusion | 0.005 | 0.985 | 24000 | 400 | 0.5 | 0.1 |
| Burgers' | 0.005 | 0.985 | 18000 | 1000 | 0.5 | 0.1 |
| N-S | 0.005 | 0.985 | 24000 | 100 | 0.5 | 0.1 |
| KdV | 0.005 | 0.985 | 12000 | 200 | 0.5 | 0.1 |
| LBM | 0.001 | 0.985 | 20000 | 2000 | 0.8 | 0.1 |

## F.4 Error measurement

We reported the root normalized errors in the main text using an RNMSE measure similar to the one defined in Eq. (12). For label trajectories $\mathbf{u}_j^{T_1:T_2}$ ($j = 0, \ldots, M$) in a dataset $\mathbf{U}$ and the forecasts $\tilde{\mathbf{u}}_j^{T_1:T_2}$, the RNMSE is calculated as:

$$\text{RNMSE} := \frac{1}{Mm(T_2 - T_1)} \sum_{t=T_1}^{T_2} \sum_{j=1}^{M} \sum_{i=1}^{m} \frac{\left\| {}_i\tilde{\mathbf{u}}_j^t - {}_i\mathbf{u}_j^t \right\|_{\mathcal{X}}}{\left\| {}_i\mathbf{u}_j^t \right\|_{\mathcal{X}}}, \tag{19}$$

where $\| \cdot \|_{\mathcal{X}}$ is the Frobenius norm over the spatial axes. As also suggested in [26], this error measure ensures that the model is evaluated for capturing the change in the scale of the solutions over time.

## F.5 Training and inference settings

**Full grid training.** All experiments in Sections 4.2 and 4.3 are conducted by training the models with data on the full solver grids, i.e. $\mathcal{X}_{tr} = \mathcal{X}_{\mathcal{S}}$. The test grids $\mathcal{X}_{te}$ in Table 1 are all equal to $\mathcal{X}_{\mathcal{S}}$ (except for ↓AD-CROM). Test grids in Table 2 are either the full solver grid or randomly sampled subsets of the solver grid with 5% and 2% of its size. In the latter sub-sampled inference case, only the initial conditions $\mathbf{u}^{t_0}$ on the sub-sampled grids are employed for the inversion, while the models still reconstruct the predicted solutions on $\mathcal{X}_{\mathcal{S}}$ and are evaluated for recovering the full field. In summary, the evaluation steps for Φ-ROM and DINo are as follows.

1. Given an initial condition $\mathbf{u}^{t_0}$ on a grid $\mathcal{X}_{te}$, find $\hat{\alpha}^0$ by inversion.

2. For the time interval $[0, T_{te}]$ (containing training and test time steps), integrate $\Psi_\phi$ starting with $\hat{\alpha}^0$ to find all the corresponding $\left[\tilde{\alpha}^0, \ldots, \tilde{\alpha}^{T_{tr}}, \ldots, \tilde{\alpha}^{T_{te}}\right]$.

3. Reconstruct the forecast latent coordinates at all time steps on grid $\mathcal{X}_{\mathcal{S}}$ to find $\left[\tilde{\mathbf{u}}^0, \ldots, \tilde{\mathbf{u}}^{T_{te}}\right]$.

4. Measure the average temporal interpolation ($\left[\tilde{\mathbf{u}}^0, \ldots, \tilde{\mathbf{u}}^{T_{tr}}\right]$) and extrapolation ($\left[\tilde{\mathbf{u}}^{T_{tr}}, \ldots, \tilde{\mathbf{u}}^{T_{te}}\right]$) errors separately (as explained in Appendix F.4) using the labels $\left[\mathbf{u}^{T_0}, \ldots, \mathbf{u}^{T_{te}}\right]$.

**Training on sparse irregular grids.** Experiments in Section 4.4 are conducted by training models on sub-sampled training grids. For each dataset, a fixed training grid $\mathcal{X}_{tr}$ is randomly sampled from $\mathcal{X}_{\mathcal{S}}$ with 5% and 2% of the original grid size (see Appendix D). This grid is fixed and consistent among all the trajectories and snapshots in a dataset. Φ-ROM and DINo are then trained with the

data only observed at the sub-sampled training locations. During testing, the models are evaluated for recovering the full grid solution fields similar to the full grid training case.

## G   On the regularizing effect of $\lambda$

The hyperparameter $\lambda$ balances the two loss terms in the training objective. Note that while the latent manifold is constructed by $L_{rec}$, it is regularized by $L_{dyn}$ in order to enforce the PDE. Consequently, smaller $\lambda$ increases the regularization effect of $L_{dyn}$ on the decoder and the latent manifold, resulting in a higher reconstruction loss. At the same time, a very large choice of $\lambda$ ($> 0.9$) may result in a disparity between the actual latent dynamics and what the dynamics network $\Psi$ learns as the latent dynamics. Thus, we suggested the 0.5 to 0.8 range for the choice of $\lambda$. A larger $\lambda$ may improve the forecast error at very early time-steps in exchange for higher error accumulation, while smaller values may deteriorate the reconstructions and result in wrong time derivatives from the solver during the training. In addition, note that as we use normalized errors for both loss functions, the losses would often be in the same order of magnitude when $\lambda$ is close to 0.5 and decrease with similar rates. Below, we further explain our choice of $\lambda$ in the case of Burgers' and LBM experiments.

### G.1   Burgers' ablation study

To better demonstrate the effect of $\lambda$, we trained Burgers' with $\lambda = 0.1, 0.5, 0.9$ and report the results in Table 7. As discussed above, increasing $\lambda$ decreases the regularization effect and thus results in smaller reconstruction loss, but fails to minimize the dynamics loss. Conversely, a small $\lambda$ increases the reconstruction error in a way that the dynamics learned from the solver may be wrong or noisy.

Table 7: Loss and error values for Burgers' with different choices of $\lambda$.

| $\lambda$ | $L_{rec}$ | $L_{dyn}$ | Error $T_{tr}$ | Error $T_{te}$ |
|---|---|---|---|---|
| 0.9 | 0.002 | 0.081 | 0.137 | 0.150 |
| 0.5 | 0.019 | 0.013 | 0.021 | 0.028 |
| 0.1 | 0.176 | 0.011 | 0.459 | 0.613 |

### G.2   LBM case

In our experiments, $\lambda = 0.5$ worked well for all problems, except for LBM, which has a more complex geometry because of the discontinuity at the cylinder. Due to the imperfect reconstruction of the cylinder, we observed noise and unexpected artifacts in the time derivatives given by the PDE solver. To alleviate this issue, we increased $\lambda$ to achieve a smaller reconstruction loss, which helped with stable training of the model and improving its generalization.

## H   Extended results

### H.1   Mean squared errors

In the main text, we reported the root normalized errors for our experiments. In cases where the scale of the solutions decays over time (Diffusion, N-S, and KdV), non-normalized error measures such as mean squared error (MSE) may indicate a decreasing error over time, while normalized errors penalize the model for not adapting to the changing scale of the solutions and indicate a growing error. Nevertheless, many other works report results in MSE. For completeness, the mean squared errors for all experiments and both training and test sets are reported here in Tables 8 and 9.

### H.2   Error over time

Figure 7 shows the train and test RNMSEs at every interpolation and extrapolation time step. We note that both $\Phi$-ROM and DINo fail in longer forecasting beyond $T_{te}$ for LBM.

Table 8: Similar to Table 1 in the main text but indicating mean squared errors (MSE) for all experiments.

| Time | Diffusion $[0, T_{tr}]$ | | $[T_{tr}, T_{te}]$ | | Burgers' $[0, T_{tr}]$ | | $[T_{tr}, T_{te}]$ | |
|---|---|---|---|---|---|---|---|---|
| Dataset | $\mathbf{U}_{tr}$ | $\mathbf{U}_{te}$ | $\mathbf{U}_{tr}$ | $\mathbf{U}_{te}$ | $\mathbf{U}_{tr}$ | $\mathbf{U}_{te}$ | $\mathbf{U}_{tr}$ | $\mathbf{U}_{te}$ |
| Φ-ROM | 6.0e-4 | **8.2e-4** | **2.9e-5** | **3.8e-5** | 2.5e-3 | 2.6e-3 | **9.6e-3** | **1.1e-2** |
| DINo | **3.6e-4** | 8.8e-4 | 9.4e-5 | 1.4e-4 | 2.6e-3 | 2.7e-3 | 6.4e-2 | 1.0e-1 |
| PINN-ROM | 5.3e-4 | 8.2e-4 | 6.0e-5 | 8.0e-5 | 7.3e-2 | 7.5e-2 | 1.4e+0 | 1.3e+0 |
| FD-CROM | 1.0e-3 | 1.7e-3 | 5.1e-3 | 6.3e-3 | **9.8e-6** | **1.2e-5** | 5.7e-2 | 8.1e-2 |
| AD-CROM | 4.0e-4 | 9.3e-4 | 4.5e-4 | 6.0e-4 | 1.1e-1 | 1.1e-2 | 4.2e-1 | 4.3e-2 |
| ↓AD-CROM | 2.2e-2 | 3.2e-2 | 3.2e-2 | 4.1e-2 | 8.8e-2 | 7.2e-2 | 5.6e-1 | 5.8e-1 |

### H.3 LBM error over parameters

Figure 8 shows the LBM errors for Reynolds numbers from the training range $\beta_{tr}$ and extrapolation range $\beta_{te}$. While DINo outperforms Φ-ROM in the parameter interpolation regime, it fails to extrapolate beyond the training parameters, highlighting the impact of Φ-ROM's physics-informed training regularization in generalizing to unseen dynamics. See Fig. 13 for parameter interpolation and extrapolation forecasting examples.

### H.4 Seed statistics

We trained Φ-ROM and DINo with three different random initializations (same initializations for the two models) for N-S. Average and standard deviation of RNMSEs over the three models are reported in Table 10. Both models achieve a small standard deviation and are stable w.r.t. the initialization.

### H.5 Predicting solution dynamics

Figs. 9–13 demonstrate predicted fields at multiple times based on unseen initial conditions or parameters and provide comparisons with baselines for all PDEs investigated in this study.

Table 9: Similar to Table 2 in the main text but indicating mean squared errors (MSE) for all experiments based on DINo and Φ-ROM.

| Time | N-S | | | | KdV | | | | LBM | | | |
|---|---|---|---|---|---|---|---|---|---|---|---|---|
| Dataset | $[0, T_{tr}]$ | | $[T_{tr}, T_{te}]$ | | $[0, T_{tr}]$ | | $[T_{tr}, T_{te}]$ | | $[0, T_{tr}]$ | | $[T_{tr}, T_{te}]$ | |
| | $\mathbf{U}_{tr}$ | $\mathbf{U}_{te}$ | $\mathbf{U}_{tr}$ | $\mathbf{U}_{te}$ | $\mathbf{U}_{tr}$ | $\mathbf{U}_{te}$ | $\mathbf{U}_{tr}$ | $\mathbf{U}_{te}$ | $\beta_{tr}$ | $\beta_{te}$ | $\beta_{tr}$ | $\beta_{te}$ |
| $\mathcal{X}_{tr} = \mathcal{X}_S = \mathcal{X}_{te}$ | | | | | | | | | | | | |
| Φ-ROM | 2.4e-4 | 1.7e-3 | 2.2e-4 | 2.0e-3 | 6.3e-3 | 7.0e-3 | 1.6e-2 | 1.7e-2 | 8.5e-6 | 5.0e-5 | 5.9e-5 | 1.4e-4 |
| DINo | 1.1e-4 | 1.9e-2 | 5.1e-3 | 3.0e-2 | 1.1e-3 | 2.6e-2 | 1.1e-2 | 3.6e-2 | 3.6e-7 | 5.3e-5 | 2.6e-3 | 3.4e-3 |
| $\mathcal{X}_{tr} = \mathcal{X}_S, |\mathcal{X}_{te}| = 5\%|\mathcal{X}_S|$ | | | | | | | | | | | | |
| Φ-ROM | 2.5e-4 | 1.7e-3 | 2.3e-4 | 2.0e-3 | 6.3e-3 | 7.0e-3 | 1.7e-2 | 1.7e-2 | 8.4e-6 | 5.0e-5 | 4.8e-4 | 1.8e-3 |
| DINo | 1.4e-4 | 1.9e-2 | 5.1e-3 | 3.3e-2 | 1.2e-3 | 2.6e-2 | 1.1e-2 | 3.6e-2 | 3.7e-7 | 5.3e-5 | 2.4e-3 | 3.2e-3 |
| $\mathcal{X}_{tr} = \mathcal{X}_S, |\mathcal{X}_{te}| = 2\%|\mathcal{X}_S|$ | | | | | | | | | | | | |
| Φ-ROM | 3.5e-4 | 1.8e-3 | 2.9e-4 | 2.2e-3 | 5.3e-2 | 5.5e-2 | 5.5e-2 | 5.7e-2 | 8.5e-6 | 5.0e-5 | 4.8e-4 | 1.8e-3 |
| DINo | 6.3e-4 | 2.0e-2 | 5.1e-3 | 3.2e-2 | 3.2e-2 | 4.6e-2 | 3.1e-2 | 4.2e-2 | 3.7e-7 | 5.3e-5 | 2.1e-3 | 3.0e-3 |

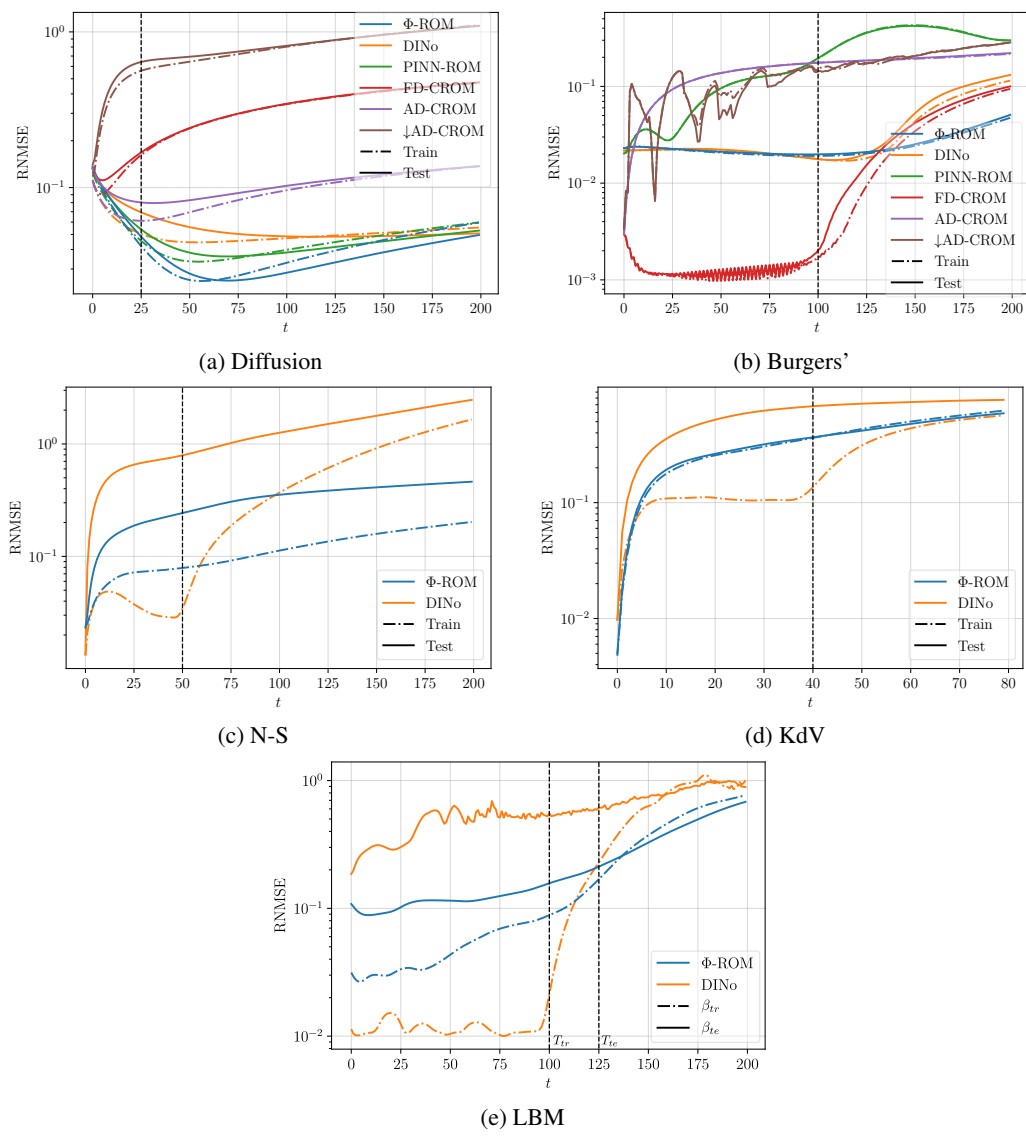

(a) Diffusion

(b) Burgers'

(c) N-S

(d) KdV

(e) LBM

Figure 7: Error accumulation over time for each dataset. Vertical dashed lines indicate $T_{tr}$. The errors for LBM are plotted beyond the test temporal horizon ($T_{te} = 125$) reported in the paper.

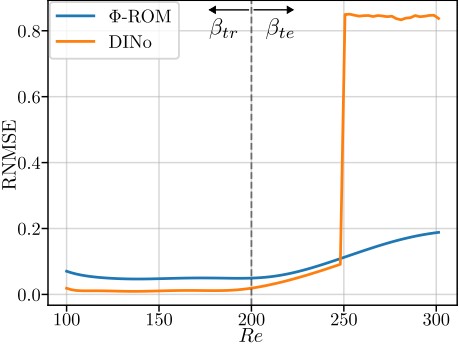

Figure 8: LBM forecasting ($[0 - T_{tr}]$) errors for Reynolds numbers sampled from the training and test intervals.

Table 10: Seed statistics of Φ-ROM and DINo trained for N-S with three different random initializations (same initializations for both models).

| Time | $[0, T_{tr}]$ | | $[T_{tr}, T_{te}]$ | |
|---|---|---|---|---|
| IC Set | $\mathbf{U}_{tr}$ | $\mathbf{U}_{te}$ | $\mathbf{U}_{tr}$ | $\mathbf{U}_{te}$ |
| | Average RNMSE | | | |
| Φ-ROM | 0.066 | 0.179 | 0.134 | 0.370 |
| DINo | 0.033 | 0.578 | 0.723 | 1.554 |
| | Standard deviation of RNMSE | | | |
| Φ-ROM | 5.6e-3 | 7.3e-3 | 1.8e-3 | 6.6e-3 |
| DINo | 1.8e-3 | 7.0e-3 | 2.5e-2 | 1.4e-2 |

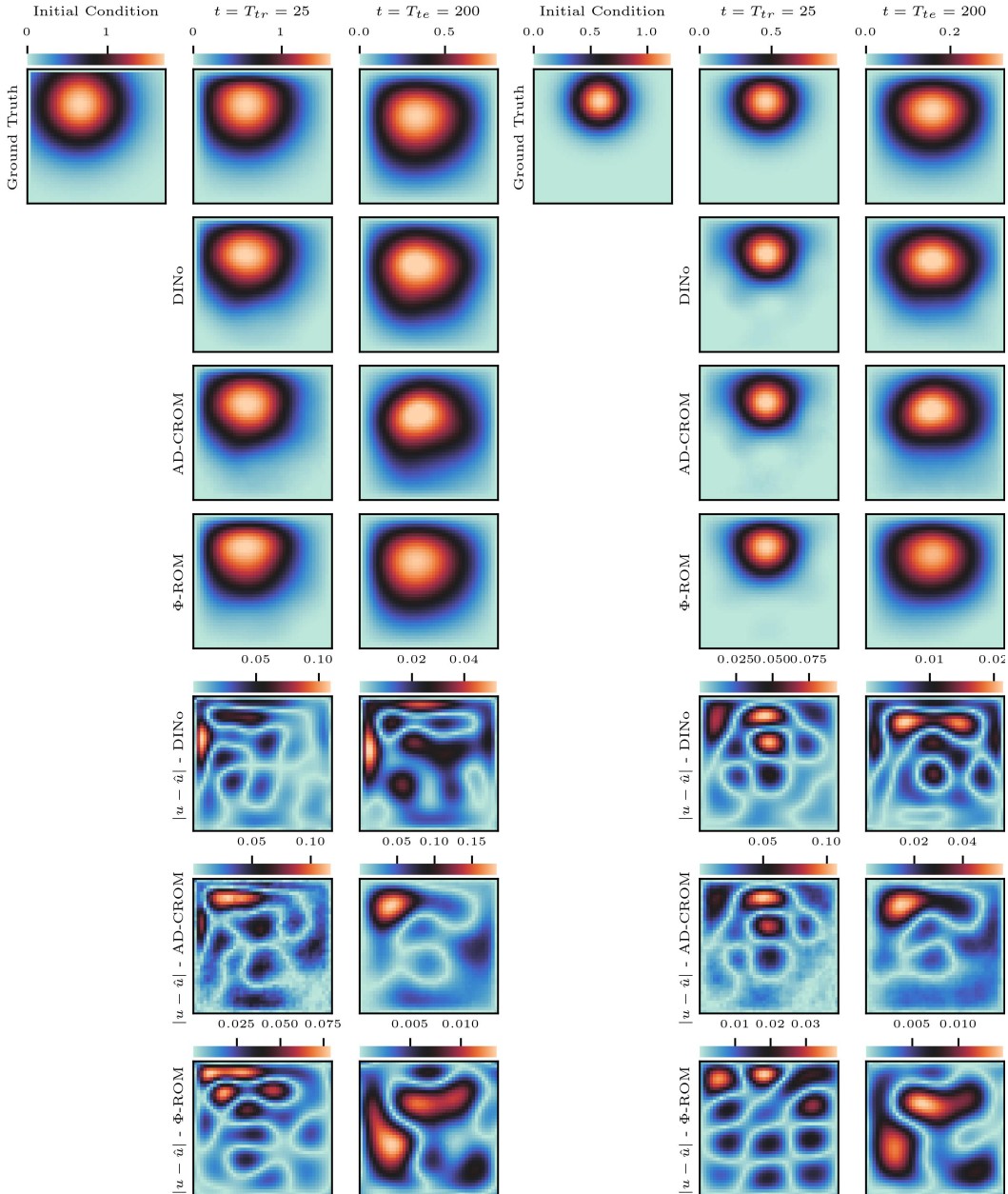

Figure 9: Predicted 2D fields for the Diffusion problem based on a test initial condition. Comparing results obtained by DINo and $\Phi$-ROM which are both trained with $N_{tr} = 100$ and $\mathcal{X}_{tr} = \mathcal{X}_{\mathcal{S}}$.

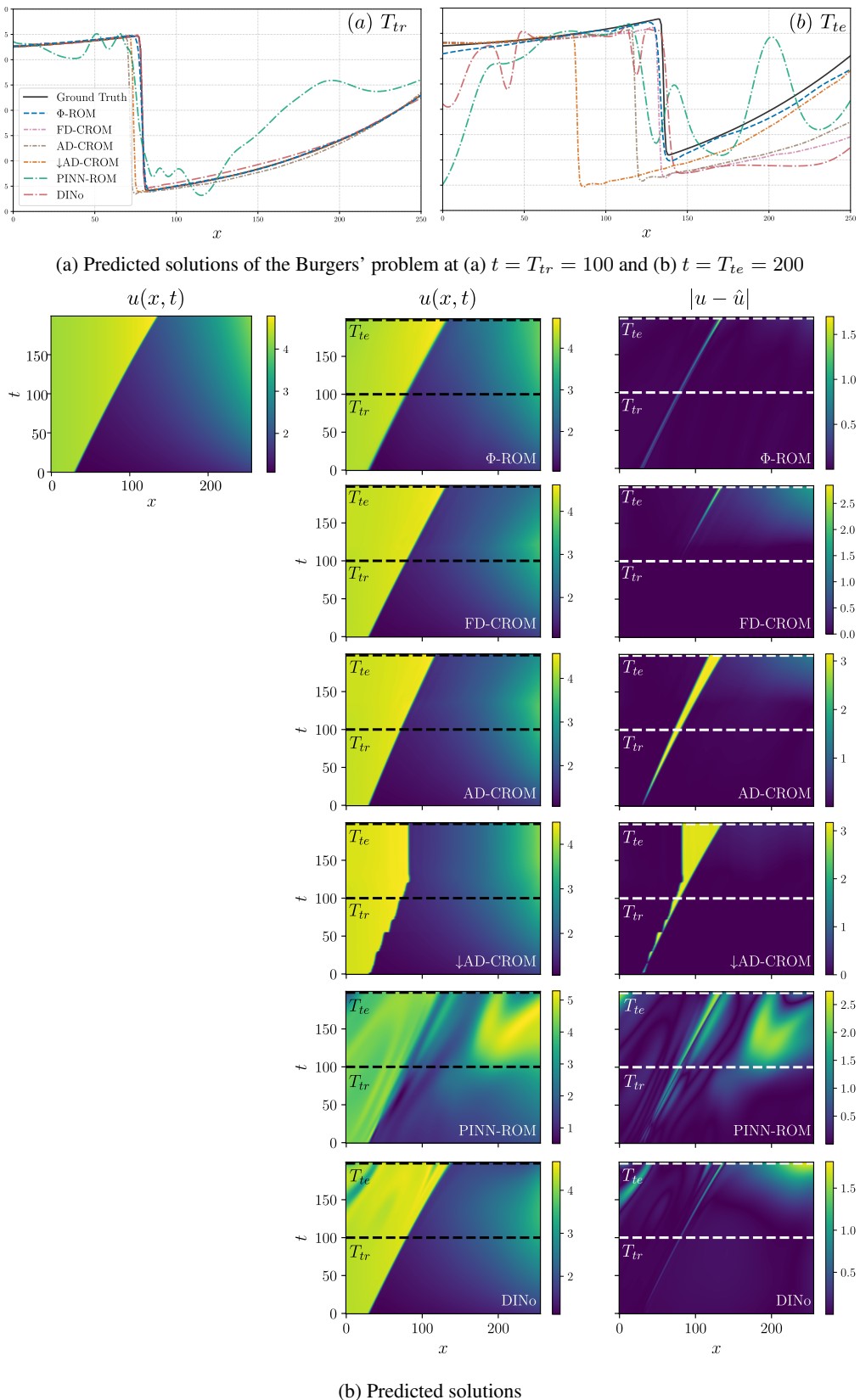

(a) Predicted solutions of the Burgers' problem at (a) $t = T_{tr} = 100$ and (b) $t = T_{te} = 200$

(b) Predicted solutions

Figure 10: Predicted dynamics of the Burgers' problem for an unseen test parameter $\mu = 0.0289$.

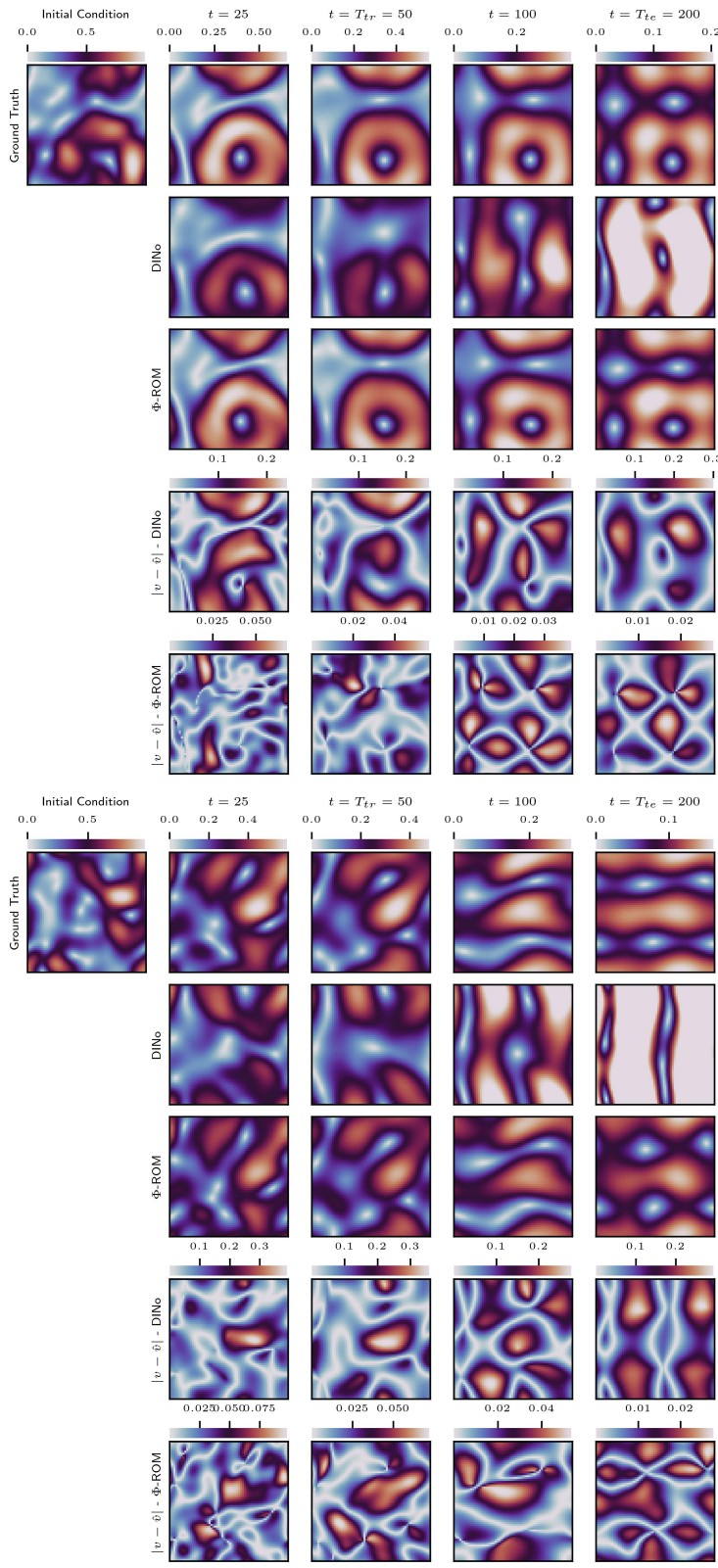

Figure 11: Predicted 2D fields for the N-S problem based on a test initial condition. Comparing results obtained by DINo and Φ-ROM which are both trained with $N_{tr} = 256$ and $\mathcal{X}_{tr} = \mathcal{X}_{\mathcal{S}}$. Plotted is the velocity magnitude and the corresponding absolute errors.

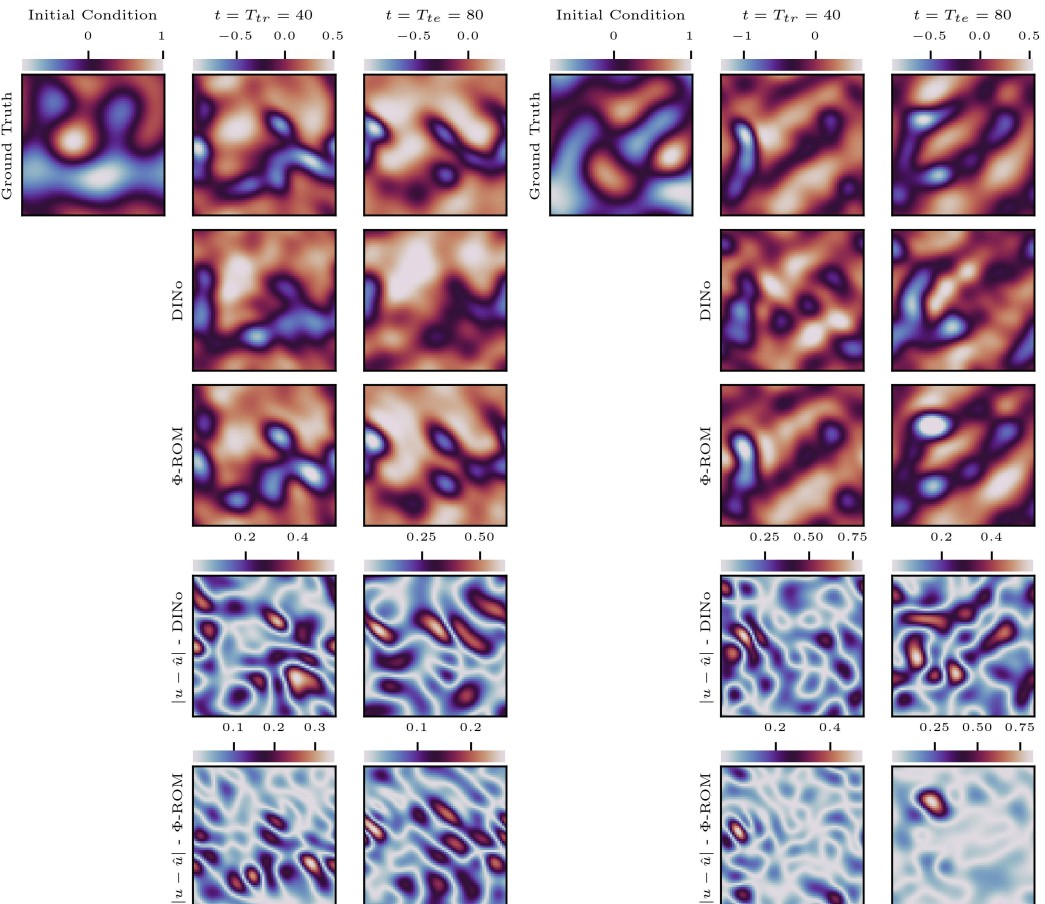

Figure 12: Predicted 2D fields for the KdV problem based on a test initial condition. Comparing results obtained by DINo and $\Phi$-ROM which are both trained with $N_{tr} = 512$ and $\mathcal{X}_{tr} = \mathcal{X}_{\mathcal{S}}$.

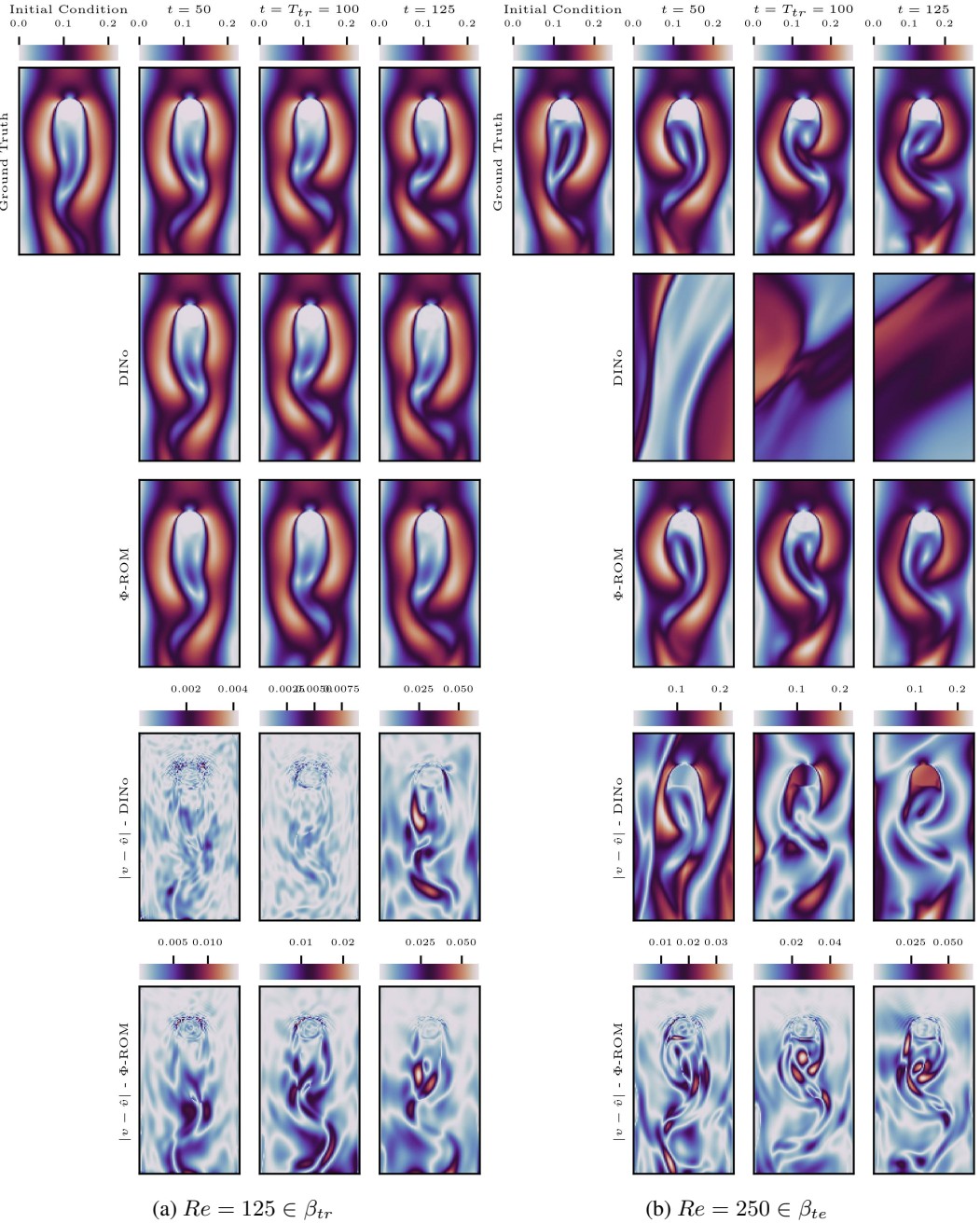

(a) $Re = 125 \in \beta_{tr}$        (b) $Re = 250 \in \beta_{te}$

Figure 13: LBM predicted dynamics for two Reynolds numbers $Re = 125$ (left panel) and $Re = 250$ (right panel) respectively from the interpolation and extrapolation parameter intervals. Comparing the magnitude of the velocity and the corresponding absolute errors obtained by DINo and $\Phi$-ROM.

