# OpenReview forum: "Physics-informed Reduced Order Modeling of Time-dependent PDEs via Differentiable Solvers"
_NeurIPS.cc/2025/Conference — NeurIPS 2025 poster_

### Official Review · Reviewer_rGnK · 2025-06-19

**Clarity:** 4
**Significance:** 2
**Originality:** 2
**Rating:** 4
**Confidence:** 5

**Summary:**

This paper presents a physics-informed framework for reduced-order modeling (ROM) of time-dependent partial differential equations (PDEs), with a focus on improving generalization and physical consistency. The authors propose Φ-ROM, a novel method that integrates differentiable PDE solvers into the learning pipeline, allowing the latent space dynamics to be directly shaped by the governing physics - this is unlike traditional data-driven ROMs, which often rely solely on high-fidelity simulations for supervision and may suffer from latent space drift. This alignment between latent and physical dynamics enables better forecasting, particularly under extrapolation to unseen initial conditions or parameter regimes. The authors demonstrate the effectiveness of their method across different PDE systems, showing improved long-term prediction accuracy, lower data requirements, and better physical interpretability.

**Questions:**

Two questions:
- The authors mentioned "When trained, the latent coordinates corresponding to a new state are computed by inversion", "The
minimization problem in Eq. (4) is solved iteratively using a gradient-based optimizer." -- why didn't the author build an encoder to construct the \alpha from u, which is probably more common in auto-encoder based ROM literatures?
- For the description of Fig 1: "The initial condition and the forecast solution at a target time T∗, can be observed and reconstructed on arbitrary (and possibly different) grids X and X′" -- why the two grids could ever be different?

**Ethical Concerns:**

["NO or VERY MINOR ethics concerns only"]

**Final Justification:**

I have revisited the CROM paper and my confusions are resolved. The authors also successfully addressed my other concerns, I'll raised my rating to reflect the changes.

**Limitations:**

Yes. The author explicitly mentioned "we acknowledge that requiring access to differentiable PDE solvers limits Φ-ROM’s immediate
applicability" which is absolutely true. And I don't think there's any major limitations for the method beyond that.

**Paper Formatting Concerns:**

No concerns.

**Quality:**

3

**Strengths And Weaknesses:**

Strengths: The paper is technically sound, with a clear motivation and rigorous formulation of the proposed Φ-ROM framework. By integrating differentiable solvers into the latent dynamics, the authors bridge a critical gap between physics-informed modeling and data-driven ROMs. The training and evaluation procedures are well-defined, and experimental results appear thorough and convincing. The paper is generally well-written and logically organized. The abstract and introduction clearly articulate the motivation and novelty. Figures and results are presented in a digestible manner.

Weakness: I have two major concerns. 1. While Φ-ROM introduces a physics-informed mechanism by incorporating differentiable solvers into the latent dynamics, it’s unclear whether this approach offers a fundamentally better alternative to PINN-ROM. PINN-ROM directly enforces the true PDEs during training—even if it performs worse empirically—its objective remains aligned with the exact continuous PDE. In contrast, Φ-ROM relies on a discretized solver whose numerical errors (local and global truncation errors, as understood from Taylor expansion analysis) mean the model is optimizing toward a discretization-dependent surrogate. This raises the concern that Φ-ROM may inherit the solver’s biases rather than learn the true physics. The paper does not sufficiently justify why this tradeoff (solver-consistency over PDE-consistency) should be preferred in principle. 2. The Φ-ROM architecture seems to fall into a regime that the original CROM paper explicitly discouraged—namely, building a low-dimensional representation (\alpha) from already discretized vector fields. CROM advocated modeling the continuous dynamics of the field itself rather than that of a discretized approximation. Yet, this paper claims Φ-ROM outperforms CROM broadly, which introduces a tension: either Φ-ROM violates the key assumptions behind CROM, or CROM’s earlier recommendations do not hold universally. The authors do not discuss this contradiction, nor clarify what might enable Φ-ROM to work better in such a regime. A clearer reconciliation between these two frameworks is warranted.

---

> ### Author Rebuttal · Authors · 2025-07-31
>
> We would like to thank the reviewer for the detailed review and insightful comments. Below, we address the questions (Q) and weaknesses (W) raised in the review.
> ## W1:
> We appreciate the reviewer’s thoughtful feedback regarding the trade-off between solver-consistency and PDE-consistency. As the reviewer is rightly aware, standard numerical solvers are designed to be consistent with the continuous PDE; they converge to the true solution as the mesh is refined and the discretization error vanishes.
>
> In our view, the fundamental issue with PINN-ROM is that it delegates the task of recovering the continuous PDE solution entirely to the neural network via automatic differentiation. In practice, this makes training significantly harder, because the network must approximate both the solution and the discretization implicitly—often without the benefit of decades of insights embedded in well-tested numerical schemes. This is one reason why PINN-ROMs (and PINNs in general) often struggle in practice, as the reviewer also observes and has pointed out in their comment.
>
> By integrating a differentiable numerical solver directly into $\Phi$-ROM, we shift this burden away from the network. The solver provides a reliable, convergent discretization of the PDE, allowing the neural network to focus its capacity on modeling the reduced latent dynamics instead of re-learning the physics from scratch. We agree that this introduces a dependency on the discretization’s properties, but we argue (as also supported by our results presented in the paper) that leveraging a well-understood, consistent solver is a principled and practical way to combine the strengths of numerical analysis and machine learning.
>
> ## W2:
> Thank you for pointing out the differences between CROM and $\Phi$-ROM. We would like to raise some points regarding CROM to help explain the apparent contradictions indicated by the reviewer between CROM and $\Phi$-ROM. First, CROM models the temporal dynamics of the PDE in the reconstructed physical space instead of the latent space. As such, a reduction in CROM is only achieved when *subsampling* is performed in the reconstructed field. Looking at the last table reported on the final page of the CROM paper, it is clear that spatial sub-sampling is absent in most of their reported experiment, meaning that inference has been on the full space, which is effectively similar to solving the full order PDE (i.e. no reduction). Second, CROM adopts two different approaches for modelling the dynamics in the reconstructed physical space: 1) A continuous model of the PDE similar to PINNs (denoted by "AD-CROM" in our paper), or 2) the discretized (e.g. finite difference-based) dynamics (denoted by "FD-CROM" in our paper). As acknowledged in the CROM paper, their computation of the network gradients using automatic differentiation for continuous physics modelling can yield erroneous gradients (as has also been studied in the PINN literature), and instead, a discretized model is used in such scenarios. It is therefore very important to note that subsampling is **not** possible (or would yield a high truncation error) when the discretized model ("FD-CROM") is used for time-stepping. As such, when CROM relies on a discretized model ("FD-CROM") due to the failure of their continuous model ("AD-CROM"), there is no subsampling reported, which implies no reduction has been achieved, and the PDE is essentially solved in a full-order manner using the discretized physics.
>
> In our experiments, we considered both versions of CROM, with and without subsampling for AD-CROM. Note that in the Burger's experiment, FD-CROM outperforms all other models for the training temporal horizon by an order of magnitude (see Table 1 and Figure 8.b). This is because FD-CROM is effectively equivalent to solving the full order PDE. However, as continuous modelling yields high gradient error near the shock, AD-CROM and its subsampled version fail to model the PDE (see Figure 3.a). Furthermore, note that CROM is trained by minimizing only the reconstruction loss of an auto-encoder. As there are no other regularizing terms, the decoder fails to generalize to unseen patterns beyond the training time horizon, which explains the relatively poor temporal extrapolation performance of FD-CROM.
>
> ## Q1:
> As the reviewer has suggested, many works use auto-*encoders*, often with fully-connected or CNN encoders. Note that in such designs, even if the decoder network is an INR and mesh-free (as in CROM), the whole encoding and decoding pipeline is **not** mesh-free because of the encoder. As such, during training, all samples must be on the same, prescribed grid $\mathcal X$ in order to be passed to the encoder, and during inference, the initial condition must again be on the same training grid $\mathcal X$ for the encoder to work and compute $\alpha^{t_0}$. Instead, by using inversion (or an auto-*decoder* similar to DINo), we have a completely mesh-free setup that does not impose any assumptions about the data grid and its consistency during training and inference. We address why this is important in our response to the next question (Q2). Nevertheless, we acknowledge that other encoder architectures, such as those based on PointNets and Transformers, may be adopted for a mesh-free setup, and we believe our physics-informed training strategy would work well with those designs too.
>
> ## Q2:
> In our experiments in Sections 4.3 and 4.4 of the paper, we demonstrated how the models can be trained and evaluated on sparse, irregular, and inconsistent observations of the input fields (e.g. the initial conditions) and still recover the full solution fields on the complete grid. In all these experiments, the input grid $\mathcal X$ and the output grid $\mathcal X'$ are different. This was possible firstly because of the mesh-free design of the network (as described in our response above to Q1), and secondly because of $\Phi$-ROM's physics-informed strategy, which enables such training with a very sparse and minimal amount of data (as compared to DINo in Tables 2 and 3 of the paper). In practice, this ability is beneficial for data assimilation tasks, especially when only sensor input observations are available, but the full solution field is required.

---

> > ### Comment · Reviewer_rGnK · 2025-08-03
> > **Response to Rebuttal**
> >
> > Thank you for your detailed response. I think most of my concerns have been clarified, I'll need to revisit the original CROM  paper to resolve the remaining confusions but that should be independent from this work here. I am still positive with this work and will keep my rating.

---

> > > ### Author Response · Authors · 2025-08-05
> > >
> > > We thank the reviewer for the response and the positive rating. We would be happy to answer any further questions regarding our work.

---

### Official Review · Reviewer_1DUk · 2025-06-23

**Clarity:** 4
**Significance:** 2
**Originality:** 3
**Rating:** 5
**Confidence:** 3

**Summary:**

The paper presents a new network-based method for reconstructing and predicting dynamics governed by spatial-temporal PDEs. It combines two network models: 1) a network that captures the solution manifold with reduced spatial dimensions; 2) a network that learns the temporal evolution of the reduced coordinates on the solution manifold. The training loss of the second network requires a differentiable PDE solver that provides temporal derivatives as supervision. The paper evaluates the proposed method on five 2D PDE problems and compares it with several standard baselines.

**Questions:**

Regarding the technical method:
1. The method’s dependence on a differentiable solver seems to arise from the dynamic loss. As Remark 3.2 suggests, one could replace this loss design with the true gradients computed from the underlying PDEs, like in PINNs. To me, this seems to be a better alternative as it avoids the dependency on a differentiable PDE solver and exploits the true time derivatives from the underlying dynamics. So, have you considered this option as suggested in Remark 3.2? Would this alternative achieve comparable or better performance than your proposed method?

2. The hyperparameter lambda is a bit unnatural and mysterious to me. Could you provide an ablation study on the effects of different lambda values and some insights for choosing them in a new problem?

3. The reconstruction and dynamic losses characterize the spatial and temporal errors, respectively. I wonder whether they could be viewed through the lens of some variational or weak forms of the underlying PDEs, which may shed more light on the physical meaning of lambda.

Regarding the experiments:

4. Are there any technical reasons that may prevent the proposed method from being applied to 3D, larger-scale PDE problems?

**Ethical Concerns:**

["NO or VERY MINOR ethics concerns only"]

**Final Justification:**

I'd like to maintain my score after reading the rebuttal and the other reviews. The rebuttal successfully addressed Questions 1-3 in my original review. The technical content in their answers to Weakness 1/Question 4 and Weakness 2 looks mostly OK, but I hope the rebuttal could fine-tune its wording to avoid overstating the paper's strengths and potential in 3D cases. Some careful text editing would be sufficient.

**Limitations:**

I appreciate that the limitation section is up front about its dependency on a differentiable solver.

I would suggest the limitation section stress that the current experiments are limited to 2D problems with small grid resolutions.

**Paper Formatting Concerns:**

None.

**Quality:**

3

**Strengths And Weaknesses:**

Strengths:
1. This paper’s writing is pretty good. In particular, I like the background and method sections, which achieve a good balance between telling the high-level story and explaining the low-level technical details.
2. The design of the network method is nice and clean. There are a number of ways of incorporating neural networks into reduced-order modeling, and I feel the overall design of the two network models provides an intuitive way of capturing the reduced-order dynamics. It is also a fairly general framework that can be applied to various PDEs and discretization schemes.

Weaknesses:
1. The experiments only consider two-dimensional PDE examples with low-resolution discretization schemes, i.e., roughly a hundred by a hundred cells according to the appendix.
2. The experiments seem to lack comparisons with numerical PDE solvers. The motivation and benefits of applying neural physics over using existing numerical solvers are unclear to me.

---

> ### Author Rebuttal · Authors · 2025-07-31
>
> We would like to thank the reviewer for the feedback and the insightful suggestions. Below, we address the questions (Q) and weaknesses (W) raised in the review.
>
> ## Q1:
> We did compare $\Phi$-ROM with two other physics-informed ROMs, specifically PINN-ROM (as defined in Remark 3.2) and CROM, in Section 4.2, and the results are reported in Table 1 of the paper. As correctly pointed out, using a PINN-inspired loss would avoid the requirement for a differentiable solver. However, as also thoroughly studied in the PINN literature, PINN loss suffers from various failure scenarios when solving PDEs with high-order terms, non-linearity, or stiff source terms [1]. As shown in Table 1, $\Phi$-ROM and PINN-ROM achieve similar performance in the linear Diffusion problem. However, as shown in Figure 3, PINN-ROM falls short in the non-linear Burger's equation and fails to model the shock. This behaviour is in line with typical failure modes of PINNs reported in the literature.
>
> [1] Krishnapriyan, Aditi, et al. "Characterizing possible failure modes in physics-informed neural networks." Advances in neural information processing systems 34 (2021): 26548-26560.
>
> ## Q2:
> The hyperparameter $\lambda$ balances the two loss terms in the training objective. Note that while the latent manifold is constructed by $L_{rec}$, it is regularized by $L_{dyn}$ in order to enforce the PDE. Consequently, smaller $\lambda$ increases the regularization effect of $L_{dyn}$ on the decoder and the latent manifold, resulting in a higher reconstruction loss. At the same time, a very large choice of lambda (>0.9) may result in a disparity between the actual latent dynamics and what the dynamics network $\Psi$ learns as the latent dynamics. Thus, we suggested the 0.5 to 0.8 range for the choice of $\lambda$. A larger $\lambda$ may improve the forecast error at very early time-steps in exchange for high error accumulation, while smaller values may deteriorate the reconstructions and result in wrong time derivatives from the solver. We also like to note that as we use normalized errors for both loss functions, the losses would be in the same order of magnitude when $\lambda$ is close to 0.5 and decrease at a similar rate.
>
> ### LBM Case
> In our experiments, $\lambda=0.5$ worked well for all problems, except for LBM, which has a more complex geometry because of the discontinuity at the cylinder. Due to the imperfect reconstruction of the cylinder, we observed noise and unexpected artifacts in the time derivatives. To alleviate this issue, we increased $\lambda$ for a smaller reconstruction loss, which helped with training the model and improving its generalization.
>
> ### Burger's ablation study
> To better demonstrate the effect of $\lambda$, we trained Burger's with $\lambda=0.1, 0.5, 0.9$ and report the results below.
>
> | $\lambda$ | $L_{rec}$ | $L_{dyn}$ | Error $[0-T_{tr}]$ | Error $[0-T_{te}]$ |
> |-:|:-:|:-:|:-:|:-:|
> |0.9 |0.002|0.081| 0.137| 0.150|
> |0.5 |0.019|0.013| 0.021| 0.028|
> |0.1 |0.176|0.011| 0.459| 0.613|
>
> **Table 1.** Loss and error values for Burger's with different choices of $\lambda$.
>
> As explained earlier, increasing $\lambda$ decreases the regularization effect and thus results in smaller reconstruction loss, but fails to minimize the dynamics loss. Conversely, a small $\lambda$ increases the reconstruction error in a way that the dynamics learned from the solver may be wrong or noisy.
>
> ## Q3:
> We are inspired by the reviewer's thinking that points to the existing similarities between the variational form of a PDE and the training procedure of $\Phi$-ROM that incorporates a PDE residual in its dynamics loss. However, it is not immediately trivial to us how a quantity similar to $\lambda$ would appear in that variational formulation that would help with further interpretation of $\lambda$ beyond what we have provided above. We believe our coupled and joint loss identifies the PDE in a new coordinate system, one which could potentially be written by composition of the latent space and physical space in an Euler-Lagrange framework. This would be an interesting theoretical investigation for future work.
>
> ## Q4 and W1:
> We understand the scaling concerns regarding the 3D and high-resolution problems. We believe there are no technical reasons to prevent $\Phi$-ROM from being applied to 3D and larger-scale PDE problems beyond a natural need for larger compute resources. We will elaborate on this in what follows.
>
> To provide a fuller picture of computational implications of such problems, we profiled $\Phi$-ROM's training costs for a high-resolution $512\times512$ 2D KdV problem ($|\mathcal X_\mathcal S| = 2^{18}$).
>
> | PDE; Res.; $\gamma$ | Total $\nabla_\theta L_{\Phi-ROM}$ | Only $\nabla_\theta L_{rec}$ | Only $\nabla_\theta L_{dyn}$ | Least squares | PDE Solver |
> |-:|:-:|:-:|:-:|:-:|:-:|
> |KdV; $64^2$; 0.1 |3.3 ms|249 $\mu$s (7.5%)|2.8 ms (84%)|547 $\mu$s (16%)|98 $\mu$s (2.9%)|
> |KdV; $512^2$; 0.1 |102 ms|7 ms (6.8%)|79 ms (77%)|3.7 ms (3.6%)|228 $\mu$s (0.2%)|
> |KdV; $512^2$; 0.002 |16 ms|7 ms (43%)|9 ms (56%)|563 $\mu$s  (3.5%)|246 $\mu$s (1.5%)|
>
> **Table 2.** Time taken by each loss function and the operations inside the dynamics loss for a single snapshot of 2D KdV measured by JAX profiler, with the portion of total training step reported in parentheses. Note that the operations above may contain additional compute overhead, which is not reported. We used a 24GB Nvidia A5000 GPU for these measurements.
>
> - **Training time.** We note that both reconstruction loss and the dynamics loss take significantly more time with increasing grid size. While we cannot control the decoding and reconstruction time present in both loss functions, we can control the hyper-reduction factor $\gamma$. Note that by reducing $\gamma$ such that the reduced grid ($\mathcal X^\downarrow_{\mathcal S}$) size is on the same order as the reduced grid for the $64^2$ case, the compute time of $L_{dyn}$ is reduced by about 90% and solving the least sqaures takes about the same time as the low-resolution case.
>
> | PDE; Resolution; $\gamma$ | Total $\nabla_\theta L_{\Phi-ROM}$ | Only $\nabla_\theta L_{rec}$ | Only $\nabla_\theta L_{dyn}$ |
> |-:|:-:|:-:|:-:|
> |KdV; $64^2$; 0.1 |668 MB|376 MB|666 MB|
> |KdV; $512^2$; 0.1 |13106 MB|13106 MB|1780 MB|
> |KdV; $512^2$; 0.002 |2854 MB|1834 MB|1780 MB|
>
> **Table 3.** Memory consumption for each loss function for one training snapshot. Note that the required memory for optimizing the total loss is less than the sum of individual losses, as JAX compiles and optimizes the operations.
>
> - **Memory consumption.** As reported in Table 3, increasing the grid size increases the memory consumption. However, adjusting $\gamma$ significantly limits the memory consumption by $L_{dyn}$ as well.
>
> - **Adjusting $\gamma$.** As noted, reducing $\gamma$ helped significantly with the training time and memory consumption. We note that hyper-reduced grids are sampled randomly and independently for each snapshot at every epoch. As such, even for small choices of $\gamma$, the reduced grids would cover the spatial domain given sufficient training time. A lower bound for selection of $\gamma$ is considering the latent size $k$, and selecting $\gamma$ such that the size of the reduced grid is at least equal to $k$. Otherwise, the resulting least squares problem would be underdetermined.
>
> - **Scaling to 3D.** As training a 3D problem would require further hyperparameter tuning regarding the network and latent size, we are not able to train $\Phi$-ROM for such a problem in a timely manner. Still, we believe the training algorithm should work well with 3D problems with adjustment of $\gamma$, although it will require more computational resources and time compared to 2D. In addition to adjusting $\gamma$, careful profiling and optimization of the code would also be essential for efficient scaling to 3D. For instance, our two loss functions are currently computed independently from each other, which means the full field is reconstructed twice in two separate forward passes of the decoder (thus requiring two backward passes). Instead, both losses can be computed in a single forward pass of the decoder.
>
> As suggested by the reviewer, we will comment on the added computational costs associated with the 3D problems in our limitations section.
>
> ## W2:
> We would like to highlight that all our error metrics are computed by comparing the results of a reduced-order model and its full-order model (i.e. the numerical PDE solver).
>
> While numerical solvers offer accurate solutions with convergence guarantees, their application in scenarios that require real-time computation of the solutions is not practical, especially when dealing with high-resolution and 3D problems. Reduced-order models try to solve the computational burden of full-order solvers by operating in a low-dimensional solution manifold. Finally, neural ROMs construct non-linear manifolds by using neural networks, which is essential for modelling of nonlinear PDEs.
>
> Compared to a numerical solver, $\Phi$-ROM operates in a fixed-size latent space. While the computational cost of a full-order numerical solver would rapidly increase with a high-resolution mesh or 3D problems, the inference speed of $\Phi$-ROM depends on the significantly smaller latent size.
>
> To provide a better picture of the difference between time-stepping in the physical space and latent space, we measured the integration time for 100 time-steps of the LBM problem using the numerical solver and $\Phi$-ROM, both running on a GPU. $\Phi$-ROM took **1.48 ms** to forecast the latent solution for 100 steps, while solving in the physical space took **19.9 ms**. Note that as the dimension of the problem grows, the gap between the reduced and full order models would grow as well.

---

> > ### Comment · Reviewer_1DUk · 2025-08-03
> > **Rebuttal response**
> >
> > Thank you for your rebuttal. I am still happy with this work and will maintain my positive score.
> >
> > I suggest the rebuttal tone down the answers to Q4/W1 until the paper can comprehensively evaluate the method on multiple 3D cases. Similarly, the answer to W2 seems overly optimistic to me unless a thorough comparison with numerical solvers on multiple test problems accompanies it. It is a bit confusing to claim that numerical methods' "applications in scenarios that require real-time computation of the solutions is not practical, especially when dealing with high-resolution and 3D problems": First, this statement may not be correct, depending on your definitions of "high-resolution" and the physics models; Second, isn't "high-resolution and 3D problems" also a limitation of your approach in the paper's current form?
> >
> > I fully understand the rush during the short rebuttal period, so not having a 3D example or not having a thorough comparison with numerical methods won't push me to reject this paper, as long as the paper can fairly and comprehensively discuss them.

---

> > > ### Author Response · Authors · 2025-08-05
> > >
> > > We thank the reviewer for their response and positive rating.
> > >
> > > We fully understand the reviewer's concern regarding 3D problems. We absolutely agree with the reviewer that any publishable claims regarding 3D problems require a comprehensive study and evaluation. We also note that training $\Phi$-ROM in its current form for 3D problems is **not** computationally cheap. Thus, as was rightly suggested by the reviewer, we will acknowledge these limitations in the revised version of the paper.
> > >
> > > We just like to clarify that our intention in providing a preliminary analysis of the computational costs and bottlenecks of $\Phi$-ROM was merely to address the fair comments raised by the reviewers, and not to make bold and finalized assessments that would appear in our published paper. In fact, the exchanged ideas during this rebuttal have motivated us to focus on a comprehensive study as our future research to demonstrate how $\Phi$-ROM may provide tractable physics-informed ROMs for 3D and large-scale problems, addressing the current caveats of our current paper.

---

> > > > ### Comment · Reviewer_1DUk · 2025-08-07
> > > > **Official Comment by Reviewer**
> > > >
> > > > Thank you for the reply. This is good to hear. I have no further questions.

---

### Official Review · Reviewer_Sn8d · 2025-06-30

**Clarity:** 2
**Significance:** 3
**Originality:** 2
**Rating:** 4
**Confidence:** 3

**Summary:**

The paper proposes a Physics-informed Reduced-Order Model that builds on the DIno framework and directly trains on latent dynamics using the labels from the differentiable PDE solver. Numerical results show the advantages of this method compared to DIno and other physics-informed ROMs.

**Questions:**

Q1) How well does the method perform compared to non-ROM methods? Including some non-ROM baselines (like Physics-informed neural operator [1] and Physics-informed deep operator network [2]) may help readers get a more comprehensive sense of the position of the proposed method in the broader field of physics-informed machine learning.

[1] Li, Zongyi, et al. "Physics-Informed Neural Operator for Learning Partial Differential Equations"

[2] Wang, Sifan, et al. "Learning the solution operator of parametric partial differential equations with physics-informed DeepONets."

Q2) Why we observe smaller forecasting errors in $[T_{tr}, T_{te}]$ compared to in $[0, T_{tr}]$ in Diffusion case of Table 1? Typically, model predictions on the training dataset should have a smaller prediction error compared to the test dataset.

Q3) Could you further explain the reasoning for the choice of $\lambda$, which depends on the sensitivity of the differentiable solver to the noise and errors in the reconstructed fields?

**Ethical Concerns:**

["NO or VERY MINOR ethics concerns only"]

**Final Justification:**

I'd like to maintain my score after reviewing the authors' rebuttal. Regarding Q1, although the results of the FNO trained with a data-driven loss are interesting, I think for a fairer comparison with Phi-ROM, FNO should be trained with both a data-driven loss and a PDE-residual loss (i.e., PINO), since Phi-ROM also has access to the solver. Additionally, without seeing quantitative results addressing the identified weaknesses (such as the exact wall-clock time for computing $\dot{\alpha}^*$, training times for different methods, and results on the cost–accuracy trade off under different latent dimensions $k$), I would like to keep my rating.

**Limitations:**

Yes, adequately described in the paper.

**Quality:**

3

**Strengths And Weaknesses:**

[Strengths]

S1) Numerical experiments are extensive, including five representative problems and five baselines.

S2) Comprehensive appendix with computation details, dataset description, hyperparameter setups, and visualizations.

S3) The proposed method generalizes much better compared to DINo.

[Weaknesses]

W1) The computation of $\dot{\alpha}^{\star}$ brings restrictions to the methods (requiring differential PDE solvers) and additional computation cost. The tradeoff between accuracy and computational cost for different baselines considered in this paper has not been fully discussed and compared.

W2) There is no guidance on how to choose the reduced latent dimension $k$. The error brought by hyper-reduction for solving $\dot{\alpha}^{\star}$ could be hard to quantify, resulting in learning inaccurate latent dynamics.

---

> ### Author Rebuttal · Authors · 2025-07-31
>
> We would like to thank the reviewer for the detailed review and the insightful comments. Below, we address the questions (Q) and the weaknesses (W) raised in the review.
> ## Q1:
> We greatly appreciate the reviewer’s feedback. We were hesitant to provide direct comparisons with non-ROM neural surrogates during the preparation of our submission, as we believed such a comparison would not be an apples-to-apples comparison. While it is true that both reduced-order models based on neural networks and other neural surrogates (e.g., neural operators) aim to accelerate the solution of PDEs, they do so in fundamentally different ways. The former evolves the dynamics in a reduced subspace and coordinate system, whereas the latter operates in the full space. This distinction becomes especially significant at scale, particularly for high-resolution and 3D problems.
>
> That being said, to address the comments raised by the reviewer, we have conducted new experiments to compare $\Phi$-ROM with two non-ROM neural surrogates by just focusing on the accuracy of the unrolling predictions.
>
> We trained autoregressive FNOs with one-step training [1] and pushforward training (FNO-PF) [2] for the Burger's and LBM problems reported in the paper. We used the Apebench library [3] for training the FNO networks. As shown below for both problems, $\Phi$-ROM outperforms FNO in temporal interpolation and extrapolation tasks.
>
> |            | $T_{tr}$ | $T_{te}$ |
> |-----------:|:--------:|:--------:|
> | $\Phi$-ROM |   0.021  |   0.028  |
> |        FNO-1 |   0.037  |   0.131  |
> |     FNO-PF |   0.049  |   0.089  |
>
> **Table 1.** Burger's unrolling errors for test samples. $\Phi$-ROM errors are the same as those reported in Table 1 of the paper. We used a 3-layer FNO with 32 hidden channels, 12 Fourier modes, and GELU activation.
>
> |            | $T_{tr}$ | $T_{te}$ |
> |-----------:|:--------:|:--------:|
> | $\Phi$-ROM |   0.065  |   0.150  |
> |        FNO-1 |   0.145  |   4.867  |
> |     FNO-PF |   0.234  |   3.276  |
>
> **Table 2.** LBM unrolling errors for test samples (parameter interpolation). $\Phi$-ROM errors are the same as those reported in Table 3 of the paper. We used a 4-layer FNO with 64 hidden channels, 16 Fourier modes, and GELU activation.
>
> [1] Li, Zongyi, et al. "Fourier neural operator for parametric partial differential equations." arXiv preprint arXiv:2010.08895 (2020).
>
> [2] Brandstetter, Johannes, Daniel Worrall, and Max Welling. "Message passing neural PDE solvers." arXiv preprint arXiv:2202.03376 (2022).
>
> [3] Koehler, Felix, Simon Niedermayr, and Nils Thuerey. "APEBench: A benchmark for autoregressive neural emulators of PDEs." Advances in Neural Information Processing Systems 37 (2024): 120252-120310.
>
> ## Q2:
> Thanks for pointing this out! We believe that the higher error for the training temporal range in the diffusion case is due to a higher reconstruction error at earlier time steps. Note that the initial conditions for the Diffusion equation are generated from 2D Gaussian distributions with one peak of varying location, standard deviation, and magnitude. As the PDE evolves, the blob is *diffused* and flattened over the whole domain. This diffusion process means that the decoder would have an easier job in reconstructing the solutions at later times. Similar phenomena are observed in PINNs, where the network tends to learn the low-frequency parts of the solution more easily. See spectral bias in neural networks [1] and PINNs [2].
>
> To further validate this, we perform inversion on two snapshots of the same trajectory at $T=0$ and $T=100$ to examine the reconstruction error independently from the temporal error accumulation. The reconstruction error (RNMSE) at $T=0$ is $0.025$ while it goes down to $0.004$ at $T=100$.
>
> [1] Cao, Yuan, et al. "Towards understanding the spectral bias of deep learning." arXiv preprint arXiv:1912.01198 (2019).
>
> [2] Wang, Sifan, Hanwen Wang, and Paris Perdikaris. "On the eigenvector bias of Fourier feature networks: From regression to solving multi-scale PDEs with physics-informed neural networks." Computer Methods in Applied Mechanics and Engineering 384 (2021): 113938.
>
> ## Q3:
> The hyperparameter $\lambda$ balances the two loss terms in the training objective. Note that while the latent manifold is constructed by $L_{rec}$, it is regularized by $L_{dyn}$ in order to enforce the PDE. Consequently, smaller $\lambda$ increases the regularization effect of $L_{dyn}$ on the decoder and the latent manifold, resulting in a higher reconstruction loss. At the same time, a very large choice of lambda (>0.9) may result in a disparity between the actual latent dynamics and what the dynamics network $\Psi$ learns as the latent dynamics. Thus, we suggested the 0.5 to 0.8 range for the choice of $\lambda$. A larger $\lambda$ may improve the forecast error at very early time-steps in exchange for high error accumulation, while smaller values may deteriorate the reconstructions and result in wrong time derivatives from the solver. In addition, we would like to note that as we use normalized errors for both loss functions, the losses would be in the same order of magnitude when $\lambda$ is close to 0.5 and decrease with similar rates.
>
> ### LBM Case
> In our experiments, $\lambda=0.5$ worked well for all problems, except for LBM, which has a more complex geometry because of the discontinuity at the cylinder. Due to the imperfect reconstruction of the cylinder, we observed noise and unexpected artifacts in the time derivatives. To alleviate this issue, we increased $\lambda$ for a smaller reconstruction loss, which helped with training the model and improving its generalization.
>
> ### Burger's ablation study
> To better demonstrate the effect of $\lambda$, we trained Burger's with $\lambda=0.1, 0.5, 0.9$ and report the results below.
>
> | $\lambda$ | $L_{rec}$ | $L_{dyn}$ | Error $[0-T_{tr}]$ | Error $[0-T_{te}]$ |
> |----------:|:---------:|:---------:|:--------------:|:--------------:|
> |       0.9 |   0.002   |   0.081   | 0.137          | 0.150          |
> |       0.5 |   0.019   |   0.013   | 0.021          | 0.028          |
> |       0.1 |   0.176   |   0.011   | 0.459          | 0.613          |
>
> **Table 1.** Loss and error values for Burger's with different choices of $\lambda$.
>
> As explained earlier, increasing $\lambda$ decreases the regularization effect and thus results in smaller reconstruction loss, but fails to minimize the dynamics loss. Conversely, a small $\lambda$ increases the reconstruction error in a way that the dynamics learned from the solver may be wrong or noisy.
>
> ## W1:
> We would like to highlight that the costs associated with computing $\dot \alpha^*$ and the use of solvers are one-time costs that affect only the training time. Furthermore, while $\Phi$-ROM has a higher training cost per sample compared to DINo and CROM, it greatly improves data efficiency (see Figure 4) and generalization to out-of-distribution parameters (see Table 3). We believe these properties, as well as the improved temporal extrapolation, justify the increased training cost.
>
> We acknowledge that training $\Phi$-ROM takes longer than the DINo and CROM baselines. CROM training involves only minimizing an auto-encoder reconstruction loss, which is significantly cheaper than both DINo and $\Phi$-ROM training. However, CROM moves this training cost to inference by solving the PDE in the physical space during inference. DINo, on the other hand, unrolls the latent dynamics by integration during the training, which $\Phi$-ROM avoids. Still, DINo's training is computationally cheaper than $\Phi$-ROM.
>
> For the KdV problem on a single 24GB Nvidia A5000 GPU, a single training epoch of DINo takes about 11 seconds, while $\Phi$-ROM takes about 34 seconds. We note that we used different hardware for the experiments in the paper (as noted in Appendix F.2, along with the training times of $\Phi$-ROM models).
>
> ## W2.1: Choice of latent dim $k$
> We note that apart from the complexity of the PDE, the choice of $k$ highly depends on the decoder architecture and its hyperparameters (e.g. activation function), as the latent manifold is the input to this network. As detailed in Appendix E, we used two different decoders adopted from CROM and DINo, and noticed that the Hyper decoder (from DINo) requires noticeably larger $k$. While we chose the latent dimensions close to those used in the CROM and DINo papers, we relied on a training with only the reconstruction loss (and without minimizing the dynamics loss) to determine the latent dimensions that achieve small reconstruction errors.
>
> ## W2.2: Hyperreduction error
> We understand the reviewer's concern regarding hyper-reduction. We note that hyper-reduced grids are sampled randomly and independently for each snapshot at every epoch. As such, the reduced grids would cover the dynamics of the whole spatial domain given sufficient training time.

---

> > ### Comment · Reviewer_Sn8d · 2025-08-05
> >
> > Thank you for your response. I will take it into account when finalizing my rating.

---

> > > ### Author Response · Authors · 2025-08-05
> > >
> > > We thank the reviewer for their response and the positive rating. We would be happy to answer any further questions regarding our work.

---

### Official Review · Reviewer_MXgS · 2025-07-03

**Clarity:** 3
**Significance:** 4
**Originality:** 3
**Rating:** 5
**Confidence:** 4

**Summary:**

This paper proposes \Phi-ROM, a physics-informed reduced-order modeling (ROM) framework that leverages differentiable numerical solvers to guide the learning of latent space dynamics for time-dependent and parameterized PDEs. Unlike prior ROM approaches that treat numerical solvers as mere data generators, \Phi-ROM integrates them into the training loop, ensuring that the learned latent dynamics remain consistent with the discretized governing physics.

The model architecture builds on an existing framework, using an implicit neural representation (INR) decoder and a dynamics network to evolve latent representations over time. By using automatic differentiation (AD) and hyper-reduction, \Phi-ROM achieves computational efficiency while being mesh-agnostic, allowing training and inference on different spatial discretizations. The authors demonstrate \Phi-ROM's performance on a variety of 1D and 2D PDEs using diverse solvers (finite difference, finite volume, and LBM), and show improved generalization to unseen parameters and sparse observations compared to baseline ROM and PINN-style models.

**Questions:**

- All test cases involve constant-coefficient PDEs. Can the authors demonstrate performance on PDEs with spatially varying diffusivity, anisotropic tensors, or stiff source terms?

- What is the bottleneck in extending this approach to 3D or higher-resolution simulations? Can the authors comment (or been better, demonstrate) an example in 3D (see next point)

- Given the reliance on Jacobians and inversion (even with hyper-reduction), does \Phi-ROM scale to high-dimensional (e.g., N ~ 10⁶) settings (of mesh size)? What are the memory/runtime implications?

- Can the method support adaptive spatiotemporal sampling during inference? Sampling more near high gradients, for instance?

**Ethical Concerns:**

["NO or VERY MINOR ethics concerns only"]

**Final Justification:**

This is a good paper (as evidenced by the generally positive reviews by all reviewers). The authors have also done a good job of adding additional results and clarifying concerns.
This is a good paper, and I keep my positive score.

**Limitations:**

Yes

**Quality:**

3

**Strengths And Weaknesses:**

*Strength*

- The paper contributes a new category of ROMs that use differentiable solvers, moving beyond traditional projection-based or PINN-augmented methods. The use of differentiable solvers as part of the training process addresses a key limitation of purely data-driven ROM (their divergence from physical constraints during long-time evolution).

- The coupling of hyper-reduction with latent dynamics training is non-trivial and is a potentially novel contribution in the ROM literature.

- The empirical comparisons are extensive and systematic. The authors test their model across various PDE types, solvers, and settings (dense vs. sparse, interpolation vs. extrapolation).

- The mesh-free (or rather the mesh-agnostic) property of the method is particularly appealing for scientific computing pipelines where spatial discretizations evolve over time (e.g., AMR, or moving meshes).

*Weakness*
- The set of benchmark problems is limited to relatively simple PDEs with constant coefficients and low spatial dimensionality (1D and 2D). These are useful for validating the method but fall short of demonstrating its utility in complex, real-world scenarios, which is arguably the key motivation for such physics-informed ROMs.

- While the components (INRs, differentiable solvers, latent dynamics) are not new individually, their composition in this context is original. Still, more emphasis could be placed on how this method fundamentally differs from DINo or PINN-ROM beyond training strategy.

---

> ### Author Rebuttal · Authors · 2025-07-31
>
> We thank the reviewer for the detailed review and insightful comments. Below, we address the questions (Q) and the weaknesses (W) raised in the review.
>
> ## Q1 and W1:
> Supporting parameters that are functions of space is an interesting direction for improvement. We must acknowledge that, in the current form of our $\Psi$ network, $\beta$ can be either a scalar or a fixed-size vector, and not a function of space. To extend the concept to spatially varying parameters, one may replace $W_\beta$ in the dynamics network (see Figure 6 in Appendix E.1) with an encoder, and the encoding of the parameter would then be concatenated with $\alpha$. The new parameter encoder can simply be a CNN if the parameter is always on a fixed grid, or a mesh-free network, such as transformer-based encoders or PointNets. We believe that this addition to the model should work well with the current training method; however, due to time and resource constraints, we are not able to validate these ideas for this rebuttal.
>
> ## Q2 and Q3:
> We understand the scaling concerns regarding the 3D and high-resolution problems. To provide a fuller picture of computational implications of such problems, we profiled $\Phi$-ROM's training costs for a high-resolution $512\times512$ 2D KdV problem ($|\mathcal X_\mathcal S| = 2^{18}$).
>
> | PDE; Res.; $\gamma$ | Total $\nabla_\theta L_{\Phi-ROM}$ | Only $\nabla_\theta L_{rec}$ | Only $\nabla_\theta L_{dyn}$ | Least squares | PDE Solver |
> |--------------------------:|:-----------------------------------:|:---------------:|:---------------:|:-------------:|:----------:|
> |              KdV; $64^2$; 0.1 |                 3.3 ms                 |      249 $\mu$s (7.5%)    |      2.8 ms (84%)    |     547 $\mu$s (16%)   |    98 $\mu$s (2.9%)  |
> |             KdV; $512^2$; 0.1 |                102 ms                |       7 ms (6.8%)     |      79 ms (77%)     |     3.7 ms (3.6%)   |   228 $\mu$s (0.2%)   |
> |           KdV; $512^2$; 0.002 |                 16 ms                 |       7 ms (43%)     |       9 ms (56%)     |     563 $\mu$s  (3.5%)  |   246 $\mu$s (1.5%)  |
>
> **Table 1.** Time taken by each loss function and the operations inside the dynamics loss for a single snapshot of 2D KdV measured by JAX profiler, with the portion of total training step reported in parentheses. Note that the operations above may contain additional compute overhead, which is not reported. We used a 24GB Nvidia A5000 GPU for these measurements.
>
> - **Training time.** We note that both reconstruction loss and the dynamics loss take significantly more time with increasing grid size. While we cannot control the decoding and reconstruction time present in both loss functions, we can control the hyper-reduction factor $\gamma$. Note that by reducing $\gamma$ such that the reduced grid ($\mathcal X^\downarrow_{\mathcal S}$) size is on the same order as the reduced grid for the $64^2$ case, the compute time of $L_{dyn}$ is reduced by about 90% and solving the least sqaures takes about the same time as the low-resolution case.
>
>
> | PDE; Resolution; $\gamma$ | Total $\nabla_\theta L_{\Phi-ROM}$ | Only $\nabla_\theta L_{rec}$ | Only $\nabla_\theta L_{dyn}$ |
> |--------------------------:|:-----------------------------------:|:---------------:|:---------------:|
> |              KdV; $64^2$; 0.1 |                668 MB               |      376 MB     |      666 MB     |
> |             KdV; $512^2$; 0.1 |               13106 MB              |     13106 MB    |     1780 MB     |
> |           KdV; $512^2$; 0.002 |               2854 MB               |     1834 MB     |     1780 MB     |
>
> **Table 2.** Memory consumption for each loss function for one training snapshot. Note that the required memory for optimizing the total loss is less than the sum of individual losses, as JAX compiles and optimizes the operations.
>
> - **Memory consumption.** As reported in Table 2, increasing the grid size increases the memory consumption. However, adjusting $\gamma$ significantly limits the memory consumption by $L_{dyn}$ as well.
>
> - **Adjusting $\gamma$.** As noted, reducing $\gamma$ helped significantly with the training time and memory consumption. We note that hyper-reduced grids are sampled randomly and independently for each snapshot at every epoch. As such, even for small choices of $\gamma$, the reduced grids would cover the spatial domain given sufficient training time. A lower bound for selection of $\gamma$ is considering the latent size $k$, and selecting $\gamma$ such that the size of the reduced grid is at least equal to $k$. Otherwise, the resulting least squares problem would be underdetermined.
>
> - **Scaling to 3D.** As training a 3D problem would require further hyperparameter tuning regarding the network and latent size, we are not able to train $\Phi$-ROM for such a problem in a timely manner. Still, we believe the training algorithm should work well with 3D problems with adjustment of $\gamma$, although it will require more computational resources and time compared to 2D. In addition to adjusting $\gamma$, careful profiling and optimization of the code would also be essential for efficient scaling to 3D. For instance, our two loss functions are currently computed independently from each other, which means the full field is reconstructed twice in two separate forward passes of the decoder (thus requiring two backward passes). Instead, both losses can be computed in a single forward pass of the decoder.
>
> ## Q4:
> As the reviewer has nicely highlighted, "The mesh-free (or rather the mesh-agnostic) property of the method is particularly appealing for scientific computing pipelines where spatial discretizations evolve over time (e.g., AMR, or moving meshes)." This is absolutely correct because any given training snapshot may live on an arbitrary grid. This capability remains effective at inference time as well because $\Phi$-ROM evolves the dynamics in the reduced latent space with its own discovered coordinates $\alpha$.
>
> Note that spatial sampling during inference only happens at the inversion step for computing $\alpha^0$, and since the decoder is continuous and mesh-free, it can perform both the inversion and reconstruction with any spatial discretization. Furthermore, the dynamics network (and thus $\Phi$-ROM) is also continuous in time, and in fact, we used Bogacki--Shampine's 3/2 ODE solver with adaptive step size for integration.
>
> ## W2:
> We appreciate the reviewer's recognition of the novelty of our work. To reflect on the reviewer's point about highlighting the novelties beyond the training, we would like to point out that the addition of the PDE parameters to the existing ROM framework is new in this work and was not proposed in those cited works.

---

### Official Review · Reviewer_axmu · 2025-07-03

**Clarity:** 2
**Significance:** 2
**Originality:** 2
**Rating:** 4
**Confidence:** 2

**Summary:**

This work proposes a physics-informed reduced order modeling to simulate fundamental PDEs, including Diffusion, Burgers, Navier-Stokes, KdV, and LBM equations. The approach integrates differentiable solvers into the training process to constrain the latent dynamics and ensure consistency with governing physics. The paper validates the efficacy of the proposed framework for interpolation within the training time horizon and extrapolation beyond training time on unseen initial conditions and parameters. The method is also evaluated under sparse observation settings to demonstrate its robustness.

**Questions:**

1. This work is dependent on how latent dynamics evolve over time. How could this framework be adapted or modified for the PDEs not reliant on time? Such as elliptic PDEs

2. Operator learning methods have recently shown success for extrapolation tasks. There are no comparisons with recent operator learning methods, such as convolutional neural operators. Also, a comparison with the method PDERefiner would enable to assess the potential of the proposed method in an extrapolation scenario. Could the authors comment on this and explain how Φ-ROM compares to these methods?

3. The discussion mainly focuses on accuracy for interpolation and extrapolation tasks. Since this framework evaluates trajectories over time, how does training time scale with the increasing complexity of PDEs, higher spatial dimensions, or higher-order systems?

4. The training time horizon $T_tr$ and testing time horizon $T_{ex}$ differ for each test case. What are the criteria for selecting these time intervals, and how sensitive are the results to this choice?

**Ethical Concerns:**

["NO or VERY MINOR ethics concerns only"]

**Final Justification:**

During the rebuttal phase the authors provided additional experiments comparing the proposed method with FNO variants, and the clarification on scaling, memory, and training time, which strengthen the paper. Hence, I increased the rating to 4.

**Limitations:**

Yes, discussed

**Quality:**

2

**Strengths And Weaknesses:**

Strengths:

1. The method presentation is clear, and the training procedure and architecture description are well structured.

2. The authors provided the code, which supports reproducibility and further experimentation.

3. Several PDE test cases with varying levels of complexity are considered, and several comparisons with existing approaches are provided.

Weaknesses:

1. Comparisons with methods focused on extrapolation tasks are missing. Several neural PDE solvers already address the challenge of extrapolation, such as PDE Refiner. A comparison or discussion with respect to those methods is missing in the paper.

2. Training time and computational cost of the proposed method are not discussed or benchmarked, which is essential for assessing practical feasibility.

---

> ### Author Rebuttal · Authors · 2025-07-31
>
> We would like to thank the reviewer for the constructive feedback. Below, we address the questions (Q) and the weaknesses (W) raised in the review.
> ## Q1:
> While we do believe that our fundamental idea of incorporating differentiable PDE solvers in a neural network for constructing reduced order models can be extended to PDEs with no time-dependence, our current paper is mainly focused on time-*dependent* problems. $\Phi$-ROM connects the time variance of the solution in the physical space ($d\mathbf u/dt$) with its counterpart in the latent space ($d\alpha/dt$) and aims to learn the dynamics of the latent coordinates in time in the reduced latent space. This physics-informed training strategy is hence enabled by a secondary dynamics network, where the correspondence between the solver and the latent manifold is created. However, in a neural ROM for time-*indepdent* PDEs, we imagine that the dynamics network would be replaced by a mapping between two latent spaces, e.g. one latent space corresponding to boundary conditions and another corresponding to the solution space (see CORAL [1] for a closely related work). As the notion of "temporal dynamics" is not relevant in time-independent problems, the correspondence between the solver (i.e. physics) and the latent mapping function must be created in another way. As a result, directly extending $\Phi$-ROM (in its current form) to those scenarios is not trivial to us at this point. Investigating the best strategy to make a correspondence between the differentiable solver and the latent mapping would be a very interesting direction for future research.
>
> In light of the valid point raised by the reviewer, we will address this in the future work section of our paper.
>
> [1] Serrano, Louis, et al. "Operator learning with neural fields: Tackling pdes on general geometries." Advances in Neural Information Processing Systems 36 (2023): 70581-70611.
>
> ## Q2 and W1:
> We greatly appreciate the reviewer’s feedback. We were hesitant to provide direct comparisons with non-ROM neural surrogates during the preparation of our submission, as we believed such a comparison would not be an apples-to-apples comparison. While it is true that both neural ROMs and other neural surrogates (e.g., neural operators) aim to accelerate the solution of PDEs, they do so in fundamentally different ways. The former evolves the dynamics in a reduced subspace and coordinate system, whereas the latter operates in the full space. This distinction becomes especially significant at scale, particularly for high-resolution and 3D problems.
>
> That being said, to address the comments raised by the reviewer, we have conducted new experiments to compare $\Phi$-ROM with two non-ROM neural surrogates by just focusing on the accuracy of the unrolling predictions.
> We trained autoregressive FNOs with one-step training [1] and pushforward training (FNO-PF) [2] for the Burger's and LBM problems reported in the paper. We used the Apebench library [3] for training the FNO networks. As shown below for both problems, $\Phi$-ROM outperforms FNO in temporal interpolation and extrapolation tasks.
>
> || $T_{tr}$ |$T_{te}$|
> |-:|:-:|:-:|
> | $\Phi$-ROM |0.021|0.028|
> |FNO-1 |0.037  |0.131|
> |FNO-PF |0.049|0.089|
>
> **Table 1.** Burger's unrolling errors for test samples. $\Phi$-ROM errors are the same as those reported in Table 1 of the paper. We used a 3-layer FNO with 32 hidden channels, 12 Fourier modes, and GELU activation.
>
> || $T_{tr}$ | $T_{te}$ |
> |-:|:-:|:-:|
> | $\Phi$-ROM |0.065|0.150|
> |FNO-1 |0.145|4.867|
> |FNO-PF |0.234|3.276|
>
> **Table 2.** LBM unrolling errors for test samples (parameter interpolation). $\Phi$-ROM errors are the same as those reported in Table 3 of the paper. We used a 4-layer FNO with 64 hidden channels, 16 Fourier modes, and GELU activation.
>
> [1] Li, Zongyi, et al. "Fourier neural operator for parametric partial differential equations." arXiv preprint arXiv:2010.08895 (2020).
>
> [2] Brandstetter, Johannes, Daniel Worrall, and Max Welling. "Message passing neural PDE solvers." arXiv preprint arXiv:2202.03376 (2022).
>
> [3] Koehler, Felix, Simon Niedermayr, and Nils Thuerey. "APEBench: A benchmark for autoregressive neural emulators of PDEs." Advances in Neural Information Processing Systems 37 (2024): 120252-120310.
>
> ## Q3 and W2:
> We understand the scaling concerns regarding the 3D and high-resolution problems. To provide a fuller picture of computational implications of such problems, we profiled $\Phi$-ROM's training costs for a high-resolution $512\times512$ 2D KdV problem ($|\mathcal X_\mathcal S| = 2^{18}$).
>
> | PDE; Res.; $\gamma$ | Total $\nabla_\theta L_{\Phi-ROM}$ | Only $\nabla_\theta L_{rec}$ | Only $\nabla_\theta L_{dyn}$ | Least squares | PDE Solver |
> |--:|:--:|:--:|:--:|:---:|:--:|
> |KdV; $64^2$; 0.1 |   3.3 ms  |      249 $\mu$s (7.5%)    |      2.8 ms (84%)    |     547 $\mu$s (16%)   |    98 $\mu$s (2.9%)  |
> | KdV; $512^2$; 0.1 |  102 ms|       7 ms (6.8%)     |      79 ms (77%)     |     3.7 ms (3.6%)   |   228 $\mu$s (0.2%)   |
> |           KdV; $512^2$; 0.002 |      16 ms                 |       7 ms (43%)     |       9 ms (56%)     |     563 $\mu$s  (3.5%)  |   246 $\mu$s (1.5%)  |
>
> **Table 3.** Time taken by each loss function and the operations inside the dynamics loss for a single snapshot of 2D KdV measured by JAX profiler, with the portion of total training step reported in parentheses. Note that the operations above may contain additional compute overhead, which is not reported. We used a 24GB Nvidia A5000 GPU for these measurements.
>
> - **Training time.** We note that both reconstruction loss and the dynamics loss take significantly more time with increasing grid size. While we cannot control the decoding and reconstruction time present in both loss functions, we can control the hyper-reduction factor $\gamma$. Note that by reducing $\gamma$ for the $512^2$ case such that the reduced grid ($\mathcal X^\downarrow_{\mathcal S}$) size is on the same order as the reduced grid for the $64^2$ case, the compute time of $L_{dyn}$ is reduced by about 90% and solving the least sqaures takes about the same time as the low-resolution case. Furthermore, the time taken by the dynamics loss is reduced from 84% of the total time in $64^2$ case to 56% when $\gamma=0.002$.
>
> | PDE; Resolution; $\gamma$ | Total $\nabla_\theta L_{\Phi-ROM}$ | Only $\nabla_\theta L_{rec}$ | Only $\nabla_\theta L_{dyn}$ |
> |--:|:----:|:-:|:--:|
> |KdV; $64^2$; 0.1 |668 MB|376 MB|666 MB|
> |KdV; $512^2$; 0.1 |13106 MB|13106 MB|     1780 MB     |
> |KdV; $512^2$; 0.002 |2854 MB|1834 MB     |     1780 MB     |
>
> **Table 4.** Memory consumption for each loss function for one training snapshot. Note that the required memory for optimizing the total loss is less than the sum of individual losses, as JAX compiles the operations.
>
> - **Memory consumption.** As reported in Table 4, increasing the grid size increases the memory consumption. However, adjusting $\gamma$ significantly limits the memory consumption by $L_{dyn}$ as well.
>
> - **Adjusting $\mathbf{\gamma}$.** As noted, reducing $\gamma$ helped significantly with the training time and memory consumption. We add that hyper-reduced grids are sampled independently for each snapshot at every epoch. As such, even for small choices of $\gamma$, the reduced grids would cover the spatial domain given sufficient training time. A lower bound for selection of $\gamma$ is considering the latent size $k$, and selecting $\gamma$ such that the size of the reduced grid is at least equal to $k$. Otherwise, the resulting least squares problem would be underdetermined.
>
> - **Solver and PDE complexity.** We conducted the same experiments with the same network size for the Navier-Stokes equation (2nd order PDE) and its FVM-based solver. While we observed similar time profiles, the FVM solver took 35 $\mu$s ($64^2$) and 66 $\mu$s ($512^2$), compared to the 98 $\mu$s and 228 $\mu$s of the KdV solver, respectively. These times are purely induced by the internals of each numerical solver, as we have not applied any performance improvements to them. While our training time is affected by the timing of the PDE-solver, relying on a differentiable PDE solver does not introduce a computational bottleneck for training $\Phi$-ROM as is evident in Table 3.
>
> - **Scaling to 3D.** As training a 3D problem would require further hyperparameter tuning regarding the network and latent size, we are not able to train $\Phi$-ROM for such a problem in a timely manner. Still, we believe the training algorithm should work well with 3D problems with adjustment of $\gamma$, although it will require more computational resources and time compared to 2D. In addition to adjusting $\gamma$, careful optimization of the code would also be essential for efficient scaling to 3D. For instance, our two loss functions are computed independently from each other, which means the full field is reconstructed twice in two separate forward passes (thus also requiring two backward passes) of the decoder, while they can be merged into one forward pass.
>
> ## Q4:
> While we did not follow a specific set of rules for choosing $T_{tr}$ and $T_{te}$, we made some considerations for some datasets. In the case of Navier-Stokes, for instance, the solution is decaying in the long term. As such, we chose $T_{tr}$ such that the training would cover the development of the flow without fully entering the non-turbulent decayed regime (see Figure 11 in the Appendix). In the case of LBM, $T_{tr}$ is selected such that the training data covers the development of vortex shedding in the wake, which is followed by a periodic temporal pattern. Sensitivity to the choice of $T_{tr}$ can be seen through inclusion of such transitions in the flow regime in the training data. We also noted that both $\Phi$-ROM and DINo failed in long-term temporal extrapolation for LBM, and thus $T_{te}$ covers a short range after $T_{tr}$ (see Figure 7.e, showing LBM error up to $T=200$).

---

> > ### Comment · Reviewer_axmu · 2025-08-07
> >
> > Thanks for the detailed response. The additional experiments comparing the proposed method with FNO variants, and the clarification on scaling, memory, and training time, strengthen the paper. Including a brief discussion on the trade-offs between ROM and operator learning methods, and recommendations on how the method could be adapted for time-independent PDEs would help the readers. Overall, the responses address the key concerns, so I am raising my rating to 4.

---

> > > ### Author Response · Authors · 2025-08-08
> > >
> > > Thank you for your response and the updated assessment. We're glad to hear that our rebuttal helped clarify our work and address your concerns.

---

### Decision · Program_Chairs · 2025-09-17

**Decision:**

Accept (poster)

**Comment:**

The paper introduces Φ-ROM, a physics-informed reduced-order modeling framework that integrates differentiable PDE solvers into the training loop, ensuring latent dynamics remain consistent with discretized physics. Reviewers praised the clear methodology, mesh-agnostic design, and extensive experiments across several PDEs, noting strong generalization to unseen parameters. The main limitations are the focus on relatively simple 1D/2D problems and dependence on differentiable solvers, which may hinder scalability to larger or 3D systems. Nonetheless, the consensus is that this represents a solid and original contribution to physics-informed ROMs.